# Magma-poor continent-ocean transition zones of the southern North Atlantic: a wide-angle seismic synthesis of a new frontier

J. Kim Welford[1]

[1]Department of Earth Sciences, Memorial University of Newfoundland, St. John's, NL, Canada
**Correspondence:** J. Kim Welford (kwelford@mun.ca)

**Abstract.**

Magma-poor rifted margins, and their corresponding potential zones of exhumed serpentinized mantle, represent a unique class of tectonic boundaries with enormous promise for advancing the energy transition, such as with hydrogen production, carbon sequestration, and in the search for critical minerals. In this study, a synthesis of the results from seismic refraction/wide-angle reflection profiling, and resulting velocity models, across the continent-ocean transitions of the southern North Atlantic Ocean is presented. The models are assessed and compared to understand characteristic basement types and upper mantle behaviour across the region and between conjugate margin pairs, and to calibrate how their continent-ocean transition zones (COTZ) are defined. Ultimately, this work highlights the variable nature of continent-ocean transition zones, even within the magma-poor rifted margin end-member case, and points to avenues for future research to fill the knowledge gaps that will accelerate the energy transition.

## 1 Introduction

The transition from continental to oceanic crust at rifted margins is far more complex than originally conceived during the early development of the theory of plate tectonics (Manatschal et al., 2010; Franke, 2013; Eagles et al., 2015; Péron-Pinvidic et al., 2019). What was once envisioned to simply comprise an abrupt abutting of continental crust against oceanic crust has been revealed to more often consist of variable amounts of magmatic contributions at magma-rich margins (Eldholm et al., 1989; White, 1992) and mantle serpentinization and exhumation at magma-poor margins (Boillot et al., 1980; Whitmarsh et al., 2001; Lavier and Manatschal, 2006; Reston, 2009; Gillard et al., 2019), with the latter hosting promising targets for the energy transition, such as hydrogen production (Albers et al., 2021; Liu et al., 2023; Pérez-Gussinyé et al., 2023), carbon sequestration (Goldberg et al., 2010; Schwarzenbach et al., 2013; Coltat et al., 2021), and critical mineral exploration (Hannington et al., 2017; Patten et al., 2022).

A key component of the theory of plate tectonics was founded on the concept of the Wilson Cycle (Wilson, 1966), which involves the repeated opening and closing of successive ocean basins over geological time. This concept originated based on evidence collected around the modern Atlantic Ocean (Argand, 1922; Wilson, 1966; Williams, 1984), and the southern North Atlantic Ocean in particular, which revealed that the Paleozoic Iapetus Ocean existed before the Atlantic, with rifting

having been seeded within the deformation zones from earlier Rodinian rifting and breakup as well as subsequent Appalachian-Caledonian and Variscan orogenic episodes (Thomas, 2005).

In the southern North Atlantic Ocean, seismic refraction/wide-angle reflection surveying has allowed for the construction of lithospheric-scale velocity structural models encompassing continent-ocean transition zones that enhance and expand our understanding of the Wilson Cycle and the broader theory of plate tectonics (Fig. 1). However, unlike for the northern reaches of the North Atlantic Ocean (Funck et al., 2017), no detailed synthesis of the southern North Atlantic Ocean, with velocity models plotted at comparable scales and with comparable colour palettes, has yet been produced, despite a few regional profiles appearing in a pan-Atlantic synthesis (Biari et al., 2021). In this contribution, I present a review of seismic refraction/wide-angle reflection velocity models across the continent-ocean transition zones of magma-poor margins of the southern North Atlantic Ocean, with a view to assessing basement and upper mantle type variations, to tracking regional trends, and to reconciling continent-ocean transition zone (COTZ) definitions to a regional standard.

## 2 Geological background

Opening of the modern Atlantic Ocean resulted from the breakup and dispersal of the supercontinent Pangea (Schettino and Turco, 2009; Buiter and Torsvik, 2014; Frizon de Lamotte et al., 2015; Whalen et al., 2015; Müller et al., 2016; Peace et al., 2020), which itself formed through multiple orogenies from the end of the Neoproterozoic to the Triassic (Cawood and Buchan, 2007; Stampfli et al., 2013; Chenin et al., 2015), with closure of the Palaeozoic Iapetus Ocean during the Appalachian-Caledonide Orogen (Haworth and Keen, 1979; Williams, 1984, 1995). This zone of orogenesis ultimately corresponded to the eventual locus of Mesozoic rifting of the southern North Atlantic. Palaeozoic closure of the adjacent Rheic Ocean due to collision of northern Africa with southern Europe resulted in the Variscan Orogen, which mostly affected the European margins south of Ireland (Nance et al., 2010; Kroner and Romer, 2013; Chenin et al., 2015).

For the northernmost portion of the central Atlantic (Fig. 1), which lies south of the Newfoundland-Azores Fracture Zone (NAFZ), NW-SE-oriented rifting began in the Late Triassic (∼230 Ma; Schettino and Turco (2009)) and breakup between Nova Scotia and Morocco occurred as early as ∼190 Ma (Labails et al., 2010). Northward rift migration into the southern North Atlantic Ocean between Newfoundland and Iberia, with a reoriented W-E extension direction, occurred from the Late Jurassic to Early Cretaceous, achieving breakup at ∼115 Ma (Eddy et al., 2017). A final reorientation of the rifting to SW-NE in the Late Cretaceous (de Graciansky and Poag, 1985; Tucholke et al., 1989; Hopper et al., 2006; Tucholke et al., 2007) saw the migration of rifting to between Newfoundland and Ireland and then eventually the Labrador Sea between Labrador and Greenland from 130 Ma (Roest and Srivastava, 1989), with breakup and seafloor spreading inferred to coincide with creation of magnetic chron 27 (∼62 Ma; Gradstein et al. (2012)) in the Labrador Sea (Fig. 1; Chalmers and Laursen (1995)).

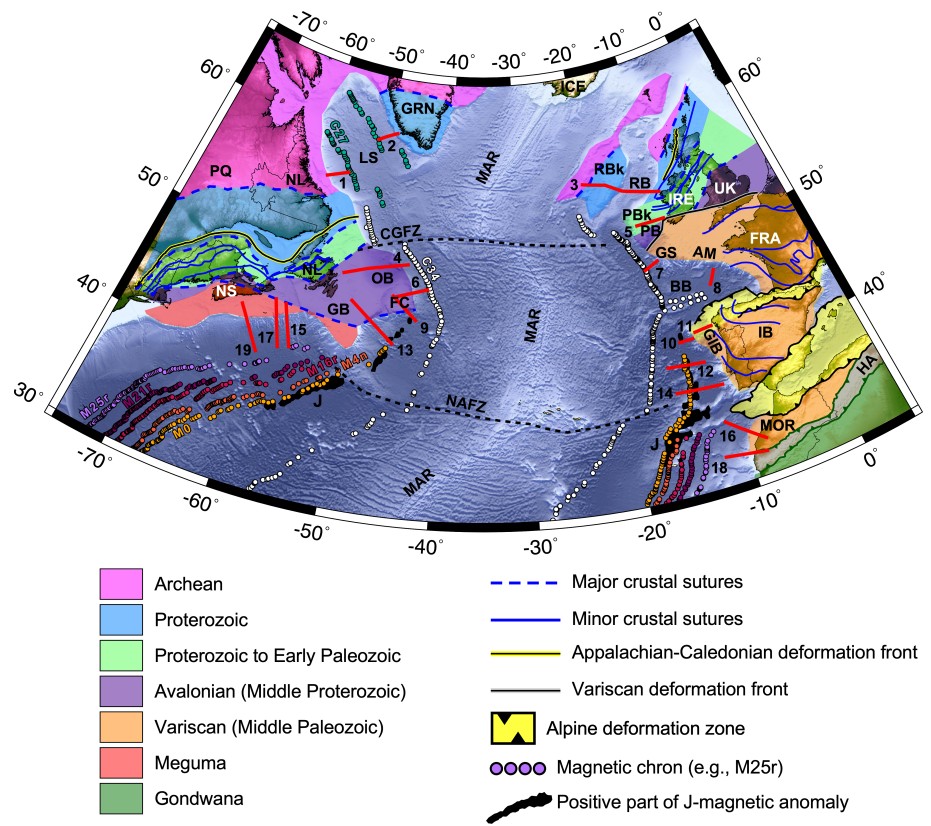

**Figure 1.** Topographic/bathymetric map from ETOPO1 (Amante and Eakins, 2009) of the study area of the southern North Atlantic Ocean subdivided by inferred basement affinity of continental crust, adapted from Jolivet et al. (2021), Tyrrell et al. (2007), Waldron et al. (2022), and Ziegler and Dèzes (2006). The numbered solid red lines correspond to the seismic refraction/wide-angle reflection (RWAR) profiles for which velocity models are reproduced in this work. The corresponding survey names and citations are provided in Table 1. Select fracture zones are delimited by dashed black lines. Select magnetic chron anomalies from Seton et al. (2014) are plotted as coloured circles (M25r, M21r, M16r, M4n, M0, C34, C27). The positive portions of the J-magnetic anomaly from Nirrengarten et al. (2016) are plotted as black polygons. Abbreviations: AM, Armorican Margin; BB, Bay of Biscay; CGFZ, Charlie-Gibbs Fracture Zone; FC, Flemish Cap; GB, Grand Banks; GIB, Galicia Interior Basin; GRN, Greenland; GS, Goban Spur; HA, High Atlas; IB, Iberia; IRE, Ireland; LS, Labrador Sea; MAR, Mid-Atlantic Ridge; MOR, Morocco; NAFZ, Newfoundland-Azores Fracture Zone; NL, Newfoundland & Labrador; NS, Nova Scotia; OB, Orphan Basin; PB, Porcupine Basin; PBk, Porcupine Bank; RB, Rockall Basin; RBk, Rockall Bank; UK, United Kingdom.

## 3 Characterization of rifted margins

Rifted margins are commonly described in terms of rift domains, which typically progress oceanward from the landward-most proximal domain, to the necking domain, to the distal domain, to the enigmatic outer domain, and ultimately to the oceanic domain (Péron-Pinvidic et al., 2013). These domains correspond to specific evolutionary stages in the rift-to-drift transition and mapping their distributions allows for direct comparison of margin structures both along strike of rifted margins and across con-

jugate margin pairs. When considering the rifted margin end-members of magma-poor versus magma-rich margins, the main

differences are typically manifested in the distal domain. Magma-poor margins in particular commonly have distal domains comprising hyperextended continental crust and potentially wide tracts of exhumed serpentinized mantle (Whitmarsh et al., 2001; Reston, 2009), displaying variable amounts of magmatic products (Bronner et al., 2011; Epin et al., 2019), while magma-rich margins have distal domains dominated by magmatic products like seaward-dipping reflectors and intrusives in the lower crust (Eldholm et al., 2000; Planke et al., 2000). Continent-ocean transition zones (COTZ) typically correspond to these distal

domains and can also include embryonic oceanic crust, accreted prior to the initiation of classic seafloor spreading (Minshull, 2009; Reston, 2009; Péron-Pinvidic et al., 2013; Sauter et al., 2018).

Magma-poor margins are the dominant margin type in the southern North Atlantic, from Nova Scotia to Labrador on the eastern Canadian margin, and from Morocco to the Irish Atlantic margin on the European side. These particular margins resulted from the propagation of rifting through ancient Appalachian-Caledonian orogenic terranes, with inheritance likely driving the

development of continental ribbons (e.g. Flemish Cap, Porcupine Bank, Rockall Bank; Péron-Pinvidic and Manatschal (2010)) and possibly setting up the necessary conditions for exhumation of serpentinized mantle within their distal domains. The development of COTZ for these margins, encompassing the final stages of rifting and the initiation of seafloor spreading, remains an area of active research, hampered by sparser data availability compared to more proximal domains, and challenged by the non-uniqueness inherent to geophysical methods in the absence of drilling. In this study, I focus on identifying the transitions from

extended continental crust (encompassing proximal, necking, and portions of the distal domains) to exhumed serpentinized mantle (a component of the distal domain), where present, to oceanic crust, whether anomalously thin, anomalously thick, or of normal thickness (5-8.5 km; White et al. (1992)). I also aim to define the lateral extents of these COTZ using a consistent classification rubric for the southern North Atlantic Ocean.

## 4    Synthesis methodology

Velocity models from twenty seismic refraction/wide-angle reflection (RWAR) surveys are included in this synthesis (see Fig. 1 and Table 1 for the profiles included in this study), from which crustal and upper mantle variations have been revealed and continent-ocean transition zones have previously been interpreted. The velocity models from the profiles highlighted in Fig. 1 are reproduced and compared in Figs. 2 to 6.

### 4.1    Geophysical methods

For studying continent-ocean transition zones, seismic RWAR techniques arguably represent the best geophysical method for determining their crustal and upper mantle velocity structures, as well as delineating fundamental lithospheric boundaries such as the Mohorivičić discontinuity (Moho), which separates the lower crust from the upper mantle. In the marine environment, these surveys involve the deployment of ocean bottom seismometers (OBS) that record arrivals from airgun sources deployed at the sea surface.

Data from RWAR surveys consist of seismic waveforms recorded as a function of time, from which traveltimes for both refracted and reflected seismic phases between the sources and the OBS can be identified and picked. The waveforms themselves contain information about the velocity variations both within layers and across subsurface interfaces. Traditionally, forward modelling using ray tracing, with localized inversion, has been used to derive velocity models from OBS traveltime data (Červený et al., 1977; McMechan and Mooney, 1980; Červený and Pšenčík, 1981, 1984; Zelt and Smith, 1992), such as for the majority of the RWAR profiles included in this study (indicated by the FMRT label in the fourth column of Table 1). More recently, tomographic inversion methods have begun to replace and complement the forward modelling approaches (Van Avendonk et al., 1998; Zelt and Barton, 1998; Korenaga et al., 2000, 2001; Van Avendonk et al., 2004; Meléndez et al., 2015; Begović, 2020). These are less time-consuming to develop and arguably less biased since the traveltimes are fit using model norms rather than subjective visual inspection. The simplest form of tomographic inversion, known as traveltime inversion (TTI), involves reproducing the picked seismic phase traveltimes (both refracted and possibly reflected), and the velocity models that were developed using this technique in the study area are indicated by the TTI label in the fourth column of Table 1. The most computationally expensive approach to modelling RWAR data involves Full-Waveform Inversion (FWI), which attempts to reproduce both the observed traveltimes as well as the seismic waveforms themselves (Pratt and Worthington, 1990; Pratt et al., 1996, 1998; Pratt, 1999; Ravaut et al., 2004; Brenders and Pratt, 2007; Virieux and Operto, 2009). As FWI requires much more closely spaced OBS than used in conventional RWAR surveys, there has not yet been widespread use of this technique offshore (Sears et al., 2008; Kamei et al., 2013; Morgan et al., 2013; Górszczyk et al., 2021), particularly in the Atlantic Ocean (Minshull and Singh, 1993; Davy et al., 2017; Guo et al., 2020; Boddupalli et al., 2021, 2022), and only one FWI-generated partial profile is considered in this synthesis ( Jian et al. (2021); line 15b in Table 1).

## 4.2 Model digitization

To allow for reproduction and direct comparison of velocity models from a range of modelling approaches and from diverse vintages of published sources, digital versions are required. Several authors graciously provided their original models as either RAYINR input files or as velocity grids, simplifying the incorporation of these models into the synthesis and ensuring accurate representations. Where original files are not available, the models are digitized directly from their published images. This is done by digitizing individual model boundaries and/or velocity contours and then manually constructing RAYINVR models from those boundaries/contours and through careful inspection of the original model images to extract velocity values as a function of model distance and depth. This process naturally introduces some uncertainty into the accuracy of the digitized model representations but great care is put into ensuring that the digital models closely resemble their analog counterparts. All of the RAYINVR files are converted to velocity grids using the VMED tool by Barry Zelt (http://www.soest.hawaii.edu/users/bzelt/vmed/vmed.html). The velocity models shown in Figs. 2 to 6 are plotted using Generic Mapping Tools (Wessel et al., 2019), the gridded velocity files, identical plot dimensions, and a common colour palette. Note that I do not have access to original OBS refraction datasets and/or the picks used to generate the presented models (other than for SIGNAL 1 (Welford et al., 2020)), so no error or uncertainty analyses can be performed. The reader is referred to the original papers (cited in Table 1) to assess the support for the individual velocity models.

## 4.3 Defining Moho

Reflections off of the Mohorivičić discontinuity (Moho) are observed as the PmP reflected phase in RWAR profiling and these are commonly used to constrain the Moho depths shown in published velocity models. For the PmP reflections to be clear during RWAR profiling, the velocity contrast between the lower crust and the upper mantle must be large enough to generate a detectable reflection. Typically, the velocity contrast across the Moho corresponds to 6.9 km/s for the lower crust and to 8 km/s for unaltered mantle, with mantle velocities decreasing with increasing degrees of serpentinization. Where mantle has been

exhumed and the crust has been removed, no Moho exists and so PmP reflections are not expected (although it is possible to observe reflections off of serpentinization fronts in the exhumed mantle domain (Gillard et al., 2019)). When forward modelling is used with discrete model boundaries (FMRT label in the fourth column of Table 1), it is not uncommon for velocities within a modelled layer to vary over such a large range as to capture both crustal velocities as well as serpentinized mantle velocities, implying that the Moho boundary should transect a model layer. This modelling oddity just means that the Moho does not

correspond to a prescribed boundary in those models. Where this phenomenon is observed in the synthesis, it is highlighted below. Where the Moho boundary in the velocity model does correspond with a clear model boundary, it is plotted with a continuous thickened black line, as is the top basement boundary where clearly defined.

## 4.4 Defining continent-ocean transition zones (COTZ)

One of the primary goals of this synthesis is to compare and contrast the COTZ along the select velocity models. The COTZ

are typically defined in terms of interpreted variations in basement type along the velocity models (e.g., continental versus transitional versus oceanic), as it is well established that for most margins, an abrupt continent-ocean boundary is not observed (Eagles et al., 2015). Transitional crust can refer to hyperextended continental crust, exhumed mantle, embryonic oceanic crust, and any combination thereof, making the demarcation of the limits of transition zones challenging. To facilitate the defining of COTZ along passive margins, Heine et al. (2013) introduced the concept of the LaLOC, the landward limit

of oceanic crust. The LaLOC marks the landward boundary between classic Penrose-type oceanic crust (Penrose Conference Participants, 1972), where normal oceanic spreading is inferred to have begun, and transition zones comprising either thinned continental crust or exhumed mantle. Sauter et al. (2023) simplify the LaLOC definition as corresponding to the landward point beyond which the top basement and Moho cease to be parallel. This latter definition does not preclude the inclusion of embryonic oceanic crust, provided it is of constant thickness. Meanwhile, another boundary often used to define COTZ is the

edge of continental crust, ECC (Nirrengarten et al., 2018). Herein, the approximate locations of the ECC and the LaLOC are indicated for each of the velocity models under consideration, although they are not used to define the full extents of the COTZ.

For this synthesis, the domain extents for basement type are extracted directly from the published interpretations. To complement the basement type comparisons herein, the lateral extents of unexhumed serpentinized mantle along each line, defined by the presence of upper mantle velocities less than 8 km/s, are also interpreted and tracked (fourth column in Table 2). This metric

allows for assessment of the lateral extent of communication between seawater and the upper mantle when the mantle has not been exhumed. To allow for direct COTZ length comparisons across the considered velocity models (fifth column in Table

2), the COTZ herein is defined as corresponding to the oceanward progression from a) hyperextended continental crust underlain by serpentinized mantle (whose landward limit would correspond to the coupling point as defined by Pérez-Gussinyé et al. (2003)), to b) exhumed serpentinized mantle (where present, and whose landward limit would be the ECC as defined by Nirrengarten et al. (2018)), and finally to c) thinner than normal oceanic crust (whose landward limit would correspond to the LaLOC according to the definition from Sauter et al. (2023)) or normal oceanic crust underlain by serpentinized mantle. The oceanward limits for the newly interpreted COTZ are marked as corresponding to the first occurence of normal thickness oceanic crust (5-8.5 km; White et al. (1992)) underlain by unaltered mantle. Based on this standardized COTZ definition, the limits of the COTZ do not necessarily correspond to the ECC and the LaLOC. Where the newly defined COTZ for each model matches that from the original RWAR publication, it is mentioned explicitly below. Finally, no COTZ are defined for failed rifts.

## 5 Geophysical constraints

### 5.1 Labrador - Greenland margins

The Labrador Sea is bordered by the rifted passive margins of Labrador in the WSW and Greenland in the ENE, with the densest concentration of RWAR surveys acquired on the Labrador margin (van der Linden, 1975; Chian et al., 1995; Reid, 1996; Funck and Louden, 1998, 1999, 2000; Funck et al., 2001a, b; Hall et al., 2002; Funck et al., 2008), although these are focused primarily on the proximal domain. While the Greenland margin has received less attention, select studies do exist (Chian and Louden, 1992; Gohl and Smithson, 1993; Chian and Louden, 1994; Funck et al., 2007) to allow a conjugate margin comparison.

Figure 2 shows the comparison of RWAR derived velocity models from either side of the Labrador Sea, resulting from the work of Chian and Louden (1994) and Chian et al. (1995) (profiles 1 and 2 in Fig. 1). These models were based on OBS data, with multi-channel (profile 1; Keen et al. (1994)) and single-channel (profile 2) seismic reflection data used to constrain the shallow sedimentary layer geometries and uppermost crust in the models. Both gravity and magnetic modelling were performed to confirm the correspondence between the derived velocity models and independent geophysical datasets (Chian et al., 1995; Chian and Louden, 1994). For inclusion of these profiles in the synthesis, the interpreted basement types are adopted directly from the original works and the upper mantle types are newly estimated from the original velocity model plots.

Landward, these models demonstrate a fundamental asymmetry, with a wider zone of thinned continental crust for the Labrador margin relative to the Greenland margin. Moving oceanward, greater symmetry is observed. On both margins, extensive sub-crustal 100 km-wide zones of serpentinized mantle are modelled, with 80 km and 70 km of those zones exhumed on the Labrador and Greenland margins, respectively. The seaward extents of both models show normal two-layered oceanic crust with a uniform boxy geometry abutting these zones of exhumation, which is consistent with their correspondence with magnetic chron 27 (Fig. 2a), interpreted as undisputed oceanic crust by Chalmers and Laursen (1995). While linear magnetic anomalies have been identified landward of chron 27 (not shown; Seton et al. (2014)), these are inferred to correspond to magnetite generated during the serpentinization process (Nazarova, 1994; Oufi et al., 2002; Sibuet et al., 2007b).

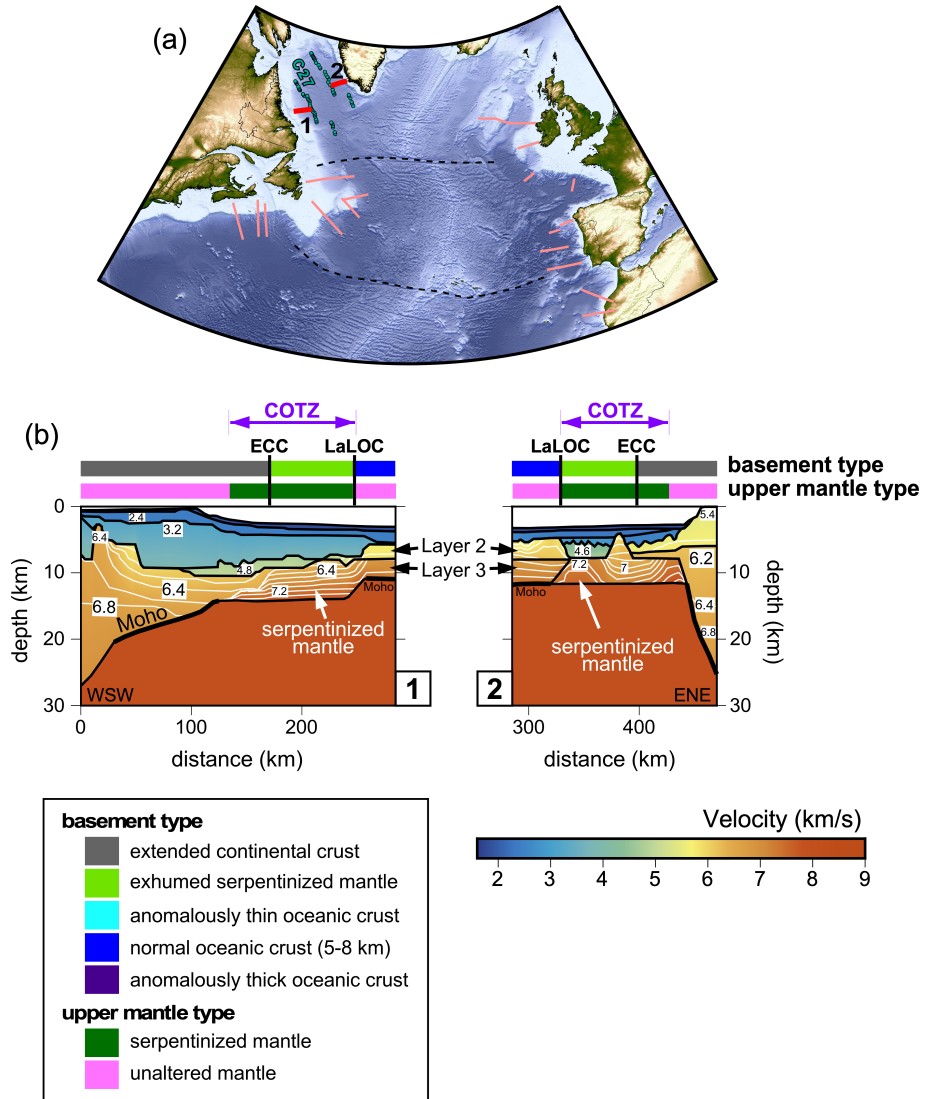

**Figure 2.** (a) Topography/bathymetry map with location of velocity models in (b) highlighted in red, and all other considered velocity model locations in this study shown in pink. (b) Velocity models reproduced from 1: Chian et al. (1995) and 2: Chian and Louden (1994). In (a), the dashed black lines are select fracture zones and magnetic chron anomaly C27 picks from Seton et al. (2014) are plotted as coloured circles. In (b), model boundaries are shown as solid black lines and velocity contours (with a contour interval of 0.2 km s$^{-1}$) are shown as solid white lines within the crust. Interpreted basement types and upper mantle types are plotted above each velocity model with colours explained in the legend. Abbreviations: COTZ, continent-ocean transition zone; ECC, edge of continental crust; LaLOC, landward limit of oceanic crust.

Note that for profile 1, the model boundary labelled as Moho in the original work (Chian et al., 1995) extends below the zone of serpentinized mantle when it should rather follow the top of the serpentinized mantle, merging with the top basement where exhumed mantle is interpreted (light green basement type). For profile 2, an oblique model boundary does separate the oceanic

layer 3 block from the zone of interpreted serpentinized mantle, while an additional model boundary delimits its bottom. For clarity, the model boundaries most consistent with a Moho definition have been plotted with thickened black lines.

The COTZ for the two Labrador Sea margins were originally defined by Chian et al. (1995) as only corresponding to the exhumed serpentinized mantle domains. In this synthesis, the COTZ are extended landward to also encompass the hyperthinned continental crust underlain by serpentinized mantle. This definition results in COTZ with lateral extents of 115 km and 100 km for profiles 1 and 2, respectively (Table 2). Note that along-strike variations in COTZ characteristics are expected in the Labrador Sea due to rheological inheritance from prior orogenesis, as modelled numerically by Gouiza and Naliboff (2021).

## 5.2 Orphan Basin - northern Irish Atlantic margins

The NE margin of Newfoundland, corresponding to the Orphan Basin, and the conjugate northern Irish Atlantic margin once lay within the thickened terranes of the Appalachian-Caledonide Orogen, with the resulting inherited structures strongly influencing subsequent rifting and the formation of multiple continental ribbons (e.g., Flemish Cap, Rockall Bank, Porcupine Bank; Fig. 1). While RWAR profiling has been undertaken in the Orphan Basin (Keen and Barrett, 1981; Chian et al., 2001; Lau et al., 2015; Welford et al., 2020), only the studies by Chian et al. (2001) and Welford et al. (2020) allowed for characterization of a portion of the COTZ (profile 4 in Fig. 1). Meanwhile, on the conjugate northern Irish Atlantic margin, there is a lack of modern RWAR studies across the continent-ocean transition, with older surveys with sparse OBS revealing simplistic abrupt continent-ocean boundaries (COB; Makris et al. (1988); Hauser et al. (1995)). As such, only inboard surveys crossing the Rockall Basin, the Porcupine Basin, and their respective flanking banks are included in this study in order to capture the velocity structures where rifting failed to achieve breakup and seafloor spreading (profiles 3 and 5 in Fig. 1).

Figure 3 shows the comparison between the velocity model from Welford et al. (2020) across the Orphan Basin (profile 4), and velocity models that lie inboard of the COTZ on the northern Irish Atlantic margin (Funck et al., 2017; Prada et al., 2017), namely profiles 3 and 5. On the NE Newfoundland margin, profile 4 was modelled from OBS data collected during the SIGNAL 2009 cruise (Funck et al., 2010) as well as the original OBS data used in Chian et al. (2001). In the resulting velocity model from Welford et al. (2020), the shallow sedimentary layer geometries were constrained using a coincident multi-channel seismic reflection profile. Gravity modelling was performed to validate the plausibility of the velocity model.

Profile 3 was acquired as part of the RAPIDS1 experiment and was originally published in Makris et al. (1991) based on modelling of OBS data alone. The southeastwardmost part of the profile was reanalysed by O'Reilly et al. (1995) and O'Reilly et al. (1996), taking into account constraints from an intersecting velocity model (Hauser et al., 1995). The velocity model for profile 3 presented herein was digitized and modified from the model in the compilation by Funck et al. (2017), which itself was based on the version of the model published by Shannon et al. (1999) for which gravity modelling was performed to support the interpreted lithospheric structures.

Profile 5 is one of three RWAR lines collected across the Porcupine Basin in 2004. This model was derived using travel-time inversion (TTI), which is performed using only the refracted and reflected phases identified on the OBS gathers (Prada et al., 2017). The velocity model was then supported through comparisons with coincident seismic reflection data, as well as gravity modelling.

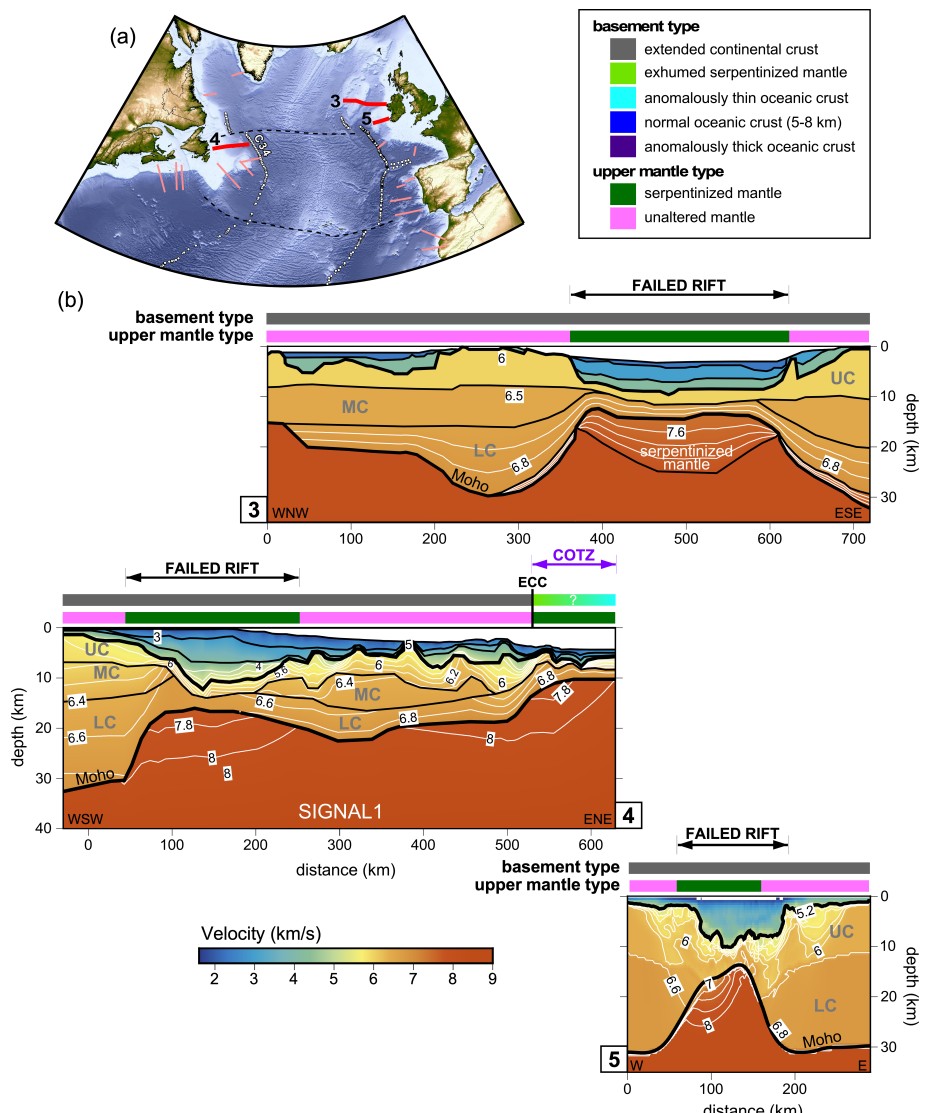

**Figure 3.** (a) Topography/bathymetry map with location of velocity models in (b) highlighted in red, and all other considered velocity model locations in this study shown in pink. (b) Velocity models reproduced from 3: Funck et al. (2017), 4: Welford et al. (2020), and 5: Prada et al. (2017). In (a), the dashed black lines are select fracture zones and magnetic chron anomaly C34 picks from Seton et al. (2014) are plotted as white circles. In (b), model boundaries are shown as solid black lines and velocity contours (with a contour interval of 0.2 km s$^{-1}$) are shown as solid white lines within the crust. Interpreted basement types and upper mantle types are plotted above each velocity model with colours explained in the legend. Abbreviations: COTZ, continent-ocean transition zone; ECC, edge of continental crust; LC, lower crust; MC, middle crust; UC, upper crust.

In terms of interpreted basement types, profiles 3, 4, and 5 are dominated by either thinned continental crust across multiple failed rifts, or by thicker continental crust in the form of continental ribbons. Across all of these failed rifts, low velocities

in the upper mantle are consistent with some degree of serpentinization (Hauser et al., 1995; O'Reilly et al., 1996; Reston et al., 2004; O'Reilly et al., 2006; Prada et al., 2017). At the COTZ, outboard of the Orphan Basin, the nature of the basement cannot be uniquely defined due to non-unique velocities but it may correspond to either exhumed mantle or anomalously thin oceanic crust, consistent with its location inboard of magnetic chron 34, the locus of undisputed oceanic crust in this part of the southern North Atlantic Ocean (Fig. 3a; Srivastava et al. (1990)). The basement types along the profiles are inherited from the published works while the mantle type lateral extents are newly interpreted herein, based on visual inspection of the original models. The extent of the COTZ (100 km; Table 2) is directly adopted from Welford et al. (2020).

## 5.3 Flemish Cap - southern Irish and Armorican margins

Southeast of the Orphan Basin on the Newfoundland margin lies the Flemish Cap, a continental ribbon that is interpreted to have been rotated and translated during Mesozoic rifting (Sibuet et al., 2007a), leading to the formation of the Orphan Basin itself. Ignoring this rotation, early studies proposed the Goban Spur, on the southern Irish margin, as the conjugate margin to the Flemish Cap (Keen and Dehler, 1993; Gerlings et al., 2012). More recent work however suggests a closer link between the Flemish Cap and the Porcupine Bank (Peace et al., 2019; Sandoval et al., 2019; Yang and Welford, 2022).

Two RWAR profiles cross the NE margin of the Flemish Cap, the 460 km-long FLAME 1 survey (profile 6 in Figs. 1 and 4; Gerlings et al. (2011)) and a shorter coincident 80 km-long experiment involving four OBS (Reid and Keen, 1990), whose results are superceded by the FLAME results. The FLAME 1 velocity model was constructed using OBS data, with a coincident seismic reflection profile used to constrain the sedimentary layer geometries. Gravity modelling was performed to confirm that the velocity model is consistent with independent geophysical constraints. In terms of basement type, the FLAME 1 velocity model is interpreted by Gerlings et al. (2011) to comprise basement with a very narrow 20 km-wide zone of exhumed serpentinized mantle, with a more expansive 80 km-wide zone of serpentinized mantle underlying thinned continental crust. An abrupt transition from exhumed mantle to normal two-layer boxy oceanic crust in the model is consistent with the intersection of the seawardmost portion of the FLAME 1 model (profile 6) with the magnetic chron 34 anomaly (Fig. 4a), the locus of undisputed oceanic crust along this portion of the southern North Atlantic (Srivastava et al., 1990). The COTZ shown for profile 6 in Fig. 4 is modified from Gerlings et al. (2011) and disregards the thinned crust underlain by unaltered mantle, resulting in a lateral extent of 80 km for the COTZ (Table 2).

On the southern Irish margin, the Goban Spur and regions immediately to the south have seen the greatest number of RWAR experiments (Ginzburg et al., 1985; Whitmarsh et al., 1986; Horsefield et al., 1994; Bullock and Minshull, 2005), with the most recent model by Bullock and Minshull (2005) corresponding to profile 7 in Figs. 1 and 4. The Bullock and Minshull (2005) model was derived using a combination of OBS and sonobuoy data (using both P and S wave arrivals, where available), constraints from a coincident seismic reflection line (WAM; Klemperer and Hobbs (1991)), velocity constraints from a nearby RWAR profile (Horsefield et al., 1994), and Deep Sea Drilling Project (DSDP) results (de Graciansky and Poag, 1985). The velocity model was ultimately used for gravity and magnetic modelling to confirm its plausibility. Based on the interpreted basement types from Bullock and Minshull (2005), the transitional crust in the model spans 120 km and is subdivided into an inboard 70 km wide zone of serpentinized mantle peridotites with subdued relief, and an oceanward 50 km wide zone

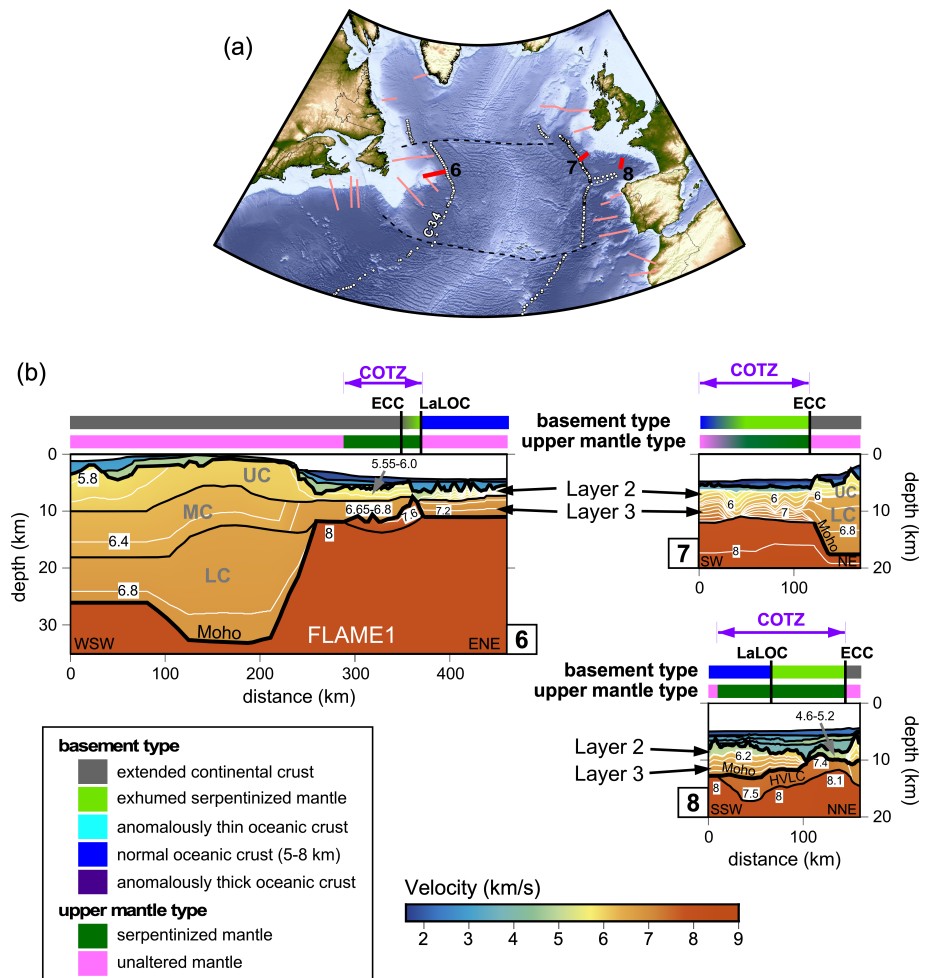

**Figure 4.** (a) Topography/bathymetry map with location of velocity models in (b) highlighted in red, and all other considered velocity model locations in this study shown in pink. (b) Velocity models reproduced from 6: Gerlings et al. (2011), 7: Bullock and Minshull (2005), and 8: Thinon et al. (2003). In (a), the dashed black lines are select fracture zones and magnetic chron anomaly C34 picks from Seton et al. (2014) are plotted as white circles. In (b), model boundaries are shown as solid black lines and velocity contours (with a contour interval of 0.2 km s$^{-1}$) are shown as solid white lines within the crust. Interpreted basement types and upper mantle types are plotted above each velocity model with colours explained in the legend. Abbreviations: COTZ, continent-ocean transition zone; ECC, edge of continental crust; HVLC, high-velocity lower crust; LaLOC, landward limit of oceanic crust; LC, lower crust; MC, middle crust; UC, upper crust.

of possibly exhumed serpentinized basement ridges, similar to ones identified on the Iberian margin to the south (Boillot et al., 1980; Beslier et al., 1993; Shipboard Scientific Party, 1998; Dean et al., 2000; Henning et al., 2004). The mantle types interpreted herein simply mirror the sections of the model with serpentinized basement. Note that this velocity model parameterization is one where the model boundary corresponding to the Moho in the NE continues southwestward, delimiting the bottom of the serpentinized mantle zone, while the Moho should actually rise steeply to meet the basement at the location

labelled ECC for edge of continental crust. For clarity, the portion of the model boundary that actually corresponds to proper Moho has been plotted with a thicker black line. Finally, the COTZ is defined according to Bullock and Minshull (2005), with a lateral extent of 120 km (Table 2).

South of the Goban Spur, along the northern margin of the Bay of Biscay lies the Armorican margin. The Norgasis 14 RWAR line (profile 8 in Fig. 1) was modelled by Thinon et al. (2003), using a coincident seismic reflection profile to help constrain the sedimentary layer geometries in the velocity model. Gravity modelling confirmed the main features of the velocity model. The basement type interpretation provided by Thinon et al. (2003) reveals an 80 km-wide zone of transitional exhumed mantle, bordered oceanward by an abrupt transition to normal thickness oceanic crust. A more extensive unexhumed sub-crustal 135 km-wide zone of serpentinized mantle was shown by Thinon et al. (2003) to extend beneath that normal oceanic crust, although its lateral extent was not explicitly stated in their work. To maintain consistency with the COTZ definition adopted herein, the oceanward limit of the COTZ is placed at the oceanward limit of the zone of unexhumed serpentinized mantle, resulting in a lateral extent of 135 km for the COTZ (Table 2).

## 5.4 Southeast Newfoundland-Iberia margins

The southeast Newfoundland and northwest Iberian margins arguably represent the best studied conjugate margin pair in the world (Louden and Chian, 1999; Srivastava et al., 2000; Manatschal et al., 2007; Péron-Pinvidic et al., 2007; Sibuet et al., 2007b; Tucholke et al., 2007; Crosby et al., 2008; Péron-Pinvidic and Manatschal, 2009; Manatschal et al., 2010; Soares et al., 2012; Péron-Pinvidic et al., 2013; Sutra et al., 2013; Mohn et al., 2015; Stanton et al., 2016; Brune et al., 2017; Eddy et al., 2017; Alves and Cunha, 2018; Causer et al., 2020; Liu et al., 2022). Herein, we focus on two RWAR surveys on the Newfoundland margin (profiles 9 and 13 in Fig. 1) and four profiles on the Iberian side (profiles 10, 11, 12, and 14 in Fig. 1). Profiles 10 and 11 combined provide almost continuous coverage from east to west across the Galicia Interior Basin, the outboard Galicia Bank continental ribbon, and the COTZ beyond (Fig. 5).

The two presented SCREECH profiles along the southeast Newfoundland margin (Fig. 5; Funck et al. (2003); Lau et al. (2006a)) were constructed based on refracted and reflected phases identified on OBS gathers, with coincident seismic reflection data used to constrain the shallow sedimentary layer geometries and basement morphologies (Hopper et al., 2004; Lau et al., 2006b). Gravity modelling was also performed by Funck et al. (2003) and Lau et al. (2006a) for both velocity models in order to assess their correspondence with independent geophysical data.

The SCREECH 1 and SCREECH 3 models (profiles 9 and 13) show abrupt oceanward crustal necking, with outboard thinned continental crust underlain by serpentinized mantle. Southeast of Flemish Cap, the thinned continental crust only extends for 20 km (profile 9) before being replaced by anomalously thin oceanic crust (60 km-wide zone) according to Funck et al. (2003). They also provide a classification of the upper mantle type with an 80 km-wide zone of unexhumed serpentinized mantle extending landward beneath the thinned continental crust (as reproduced in Fig. 5), which corresponds to the COTZ as defined in this synthesis ( Funck et al. (2003) instead defined a discrete continent-ocean boundary (COB)). Conversely, the SCREECH 3 profile off the Grand Banks shows a 90 km-wide zone of thinned continental crust transitioning into a 95 km-wide zone of exhumed serpentinized mantle, with an abrupt transition into normal oceanic crust overlying unaltered mantle.

The 190 km-wide unexhumed serpentinized mantle type highlighted in Fig. 5 was derived for this synthesis based on a visual inspection of the published model and is used to define the COTZ lateral extent of 190 km herein (Table 2). For profile 13, there is not a continuous model boundary that corresponds uniquely to the Moho, particularly one that delimits the upper boundary of the interpreted exhumed mantle. As such, only the unambiguous Moho boundaries in the parameterization are plotted with thickened black lines in Fig. 5.

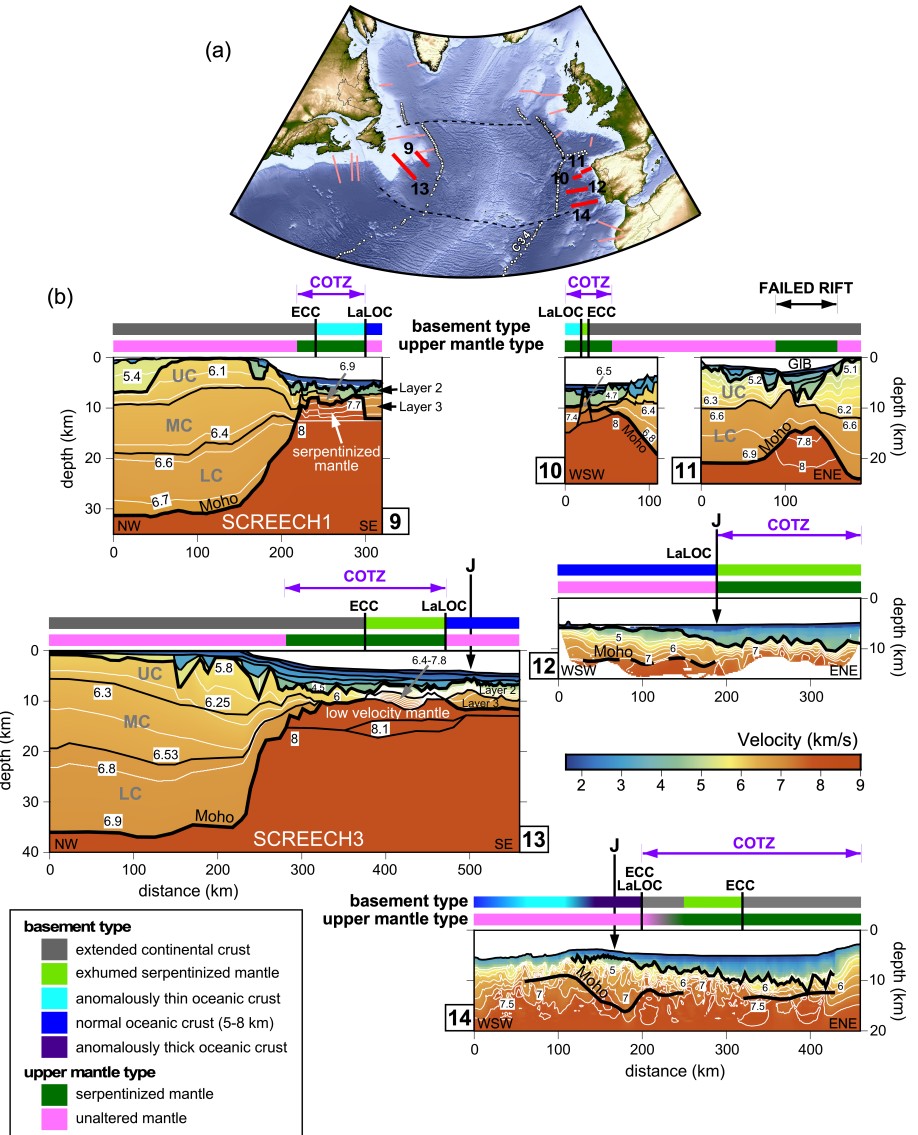

**Figure 5.** (a) Topography/bathymetry map with location of velocity models in (b) highlighted in red, and all other considered velocity model locations in this study shown in pink. (b) Velocity models reproduced from 9: Funck et al. (2003), 10: Sibuet et al. (1995) and Druet et al. (2017), 11: Pérez-Gussinyé et al. (2003), 12: Grevemeyer et al. (2022), 13: Lau et al. (2006a), and 14: Merino et al. (2021). In (a), the dashed black lines are select fracture zones and magnetic chron anomaly C34 picks from Seton et al. (2014) are plotted as white circles. In (b), model boundaries are shown as solid black lines and velocity contours (with a contour interval of 0.2 km s$^{-1}$) are shown as solid white lines within the crust. Interpreted basement types and upper mantle types are plotted above each velocity model with colours explained in the legend. The approximate location of the positive part of the J-magnetic anomaly from Nirrengarten et al. (2016) is shows for profiles 12, 13, and 14. Abbreviations: COTZ, continent-ocean transition zone; ECC, edge of continental crust; GIB, Galicia Interior Basin; LaLOC, landward limit of oceanic crust; LC, lower crust; MC, middle crust; UC, upper crust.

Across the Galicia margin, making up the northwestern Iberian margin, profile 10 (Sibuet et al., 1995; Druet et al., 2017) comprises the COTZ while profile 11 (Pérez-Gussinyé et al., 2003) is focused on the Galicia Bank continental ribbon and its inboard basin, the Galicia Interior Basin, which represents a failed rift. Profile 10 was acquired in 1987 as part of the Réframarge
cruise (cruise line 6) using only three OBS, along the track of a coincident seismic reflection profile, GP 101, which was ultimately used to constrain the sedimentary layer geometries and the basement morphology in the derived velocity model (Sibuet et al., 1995). Modelling of deep-tow magnetic data that were acquired during the RWAR campaign was also performed to enhance the interpretation of the velocity model (Sibuet et al., 1995). The version of the model used for digitization and inclusion in the synthesis is obtained from Druet et al. (2017), which was modified from Sibuet et al. (1995).

For profile 10, oceanward abrupt crustal necking of the Galicia Bank transitions to an exhumed serpentinized ridge, confirmed through drilling (Shipboard Scientific Party, 1987) and dredging (Boillot et al., 1980), and then anomalously thin oceanic crust (<5 km thick) over a further distance of approximately 20 km (Sibuet et al., 1995; Whitmarsh et al., 1996; Druet et al., 2017). The thin oceanic crust overlies a zone of serpentinized mantle (Sibuet et al., 1995) that extends 55 km landward to the westward limit of the necking zone, which is used by Druet et al. (2017) to propose the extent of the COTZ shown in Fig. 5
and adopted in this synthesis (Table 2).

Profile 11 from Pérez-Gussinyé et al. (2003) was acquired in 1997 as part of the Iberia Seismic Experiment (cruise line 17), along with a coincident seismic reflection profile, which was used to constrain sedimentary layer geometries and the top basement in the velocity model. Iterative refinement of the velocity model was achieved through its use in the prestack depth-migration of the seismic reflection profile. The basement type for all of profile 11 is interpreted as continental by Pérez-
325 Gussinyé et al. (2003). For this synthesis, lower velocities in the upper mantle are interpreted as evidence for a mantle type of unexhumed serpentinized mantle spanning 70 km based on the published velocity model.

Along the mid-Iberian margin, the velocity model for profile 12 (Figs. 1 and 5) was obtained from a joint wide-angle reflection and refraction tomographic inversion of traveltimes from densely spaced OBS (10 to 12 km spacing) along the FRAME-p3 line (Grevemeyer et al., 2022). The OBS data themselves were used for seismic mirror imaging (Grion et al.,
2007) to produce a seismic reflection section from which the sedimentary layer geometries and the top basement morphology could be determined (Grevemeyer et al., 2022). Gravity modelling was subsequently performed to validate the features in the velocity model. The model reveals basement types with an even split between exhumed serpentinized mantle landward, and normal oceanic crust oceanward. These two segments are split by the J anomaly, a prominent magnetic anomaly from the southern North Atlantic Ocean that is the subject of much controversy as to its significance (or not) as a seafloor spreading
isochron (Russell and Whitmarsh, 2003; Srivastava et al., 2000; Bronner et al., 2011; Nirrengarten et al., 2016; Stanton et al., 2016). Grevemeyer et al. (2022) conclude that it is not an isochron and so should not be used to constrain plate reconstructions but that it does mark the onset of seafloor spreading at this location along the Iberian margin. The mantle type interpreted for this synthesis mirrors the basement type split, as does the COTZ extent adopted for this synthesis (160 km; Table 2).

The Tagus Abyssal Plain within the southwest Iberian margin has been sampled by the least amount of RWAR surveys, and
340 until recently, only with widely spaced OBS (30 to 40 km spacing for Afilhado et al. (2008)). A recent velocity model (profile 14) from Merino et al. (2021), obtained using an iterative joint reflection and refraction travel-time tomography with multi-

channel seismic (MCS) constraints from a coincident 300 km-long profile, shows the highest resolution and highest degree of complexity for a model in this region, achieved using a dense OBS spacing (approximately 14 km). Gravity modelling was performed to support the interpretation of the velocity model.

For profile 14 (Fig. 5), the basement types interpreted by Merino et al. (2021) show thinned continental crust in the east abutting a 60 km-wide zone of exhumed serpentinized mantle oceanward, before a 50 km-wide rafted segment of thinned continental crust, possibly intruded by melts, is encountered. Merino et al. (2021) hypothesize that the rafted continental block was originally part of the Newfoundland margin and was orphaned on the Iberian margin due to an abrupt westward jump in the localization of extension. Immediately oceanward of the rafted continental block, a 60 km-wide zone of anomalously thick

oceanic crust is revealed, transitioning to an 80 km-wide zone of anomalously thin oceanic crust, and finally normal oceanic crust. The J-magnetic anomaly lies within the zone of thickened oceanic crust and is interpreted to correspond to the onset of seafloor spreading, as was also interpreted to the north along profile 12 (Grevemeyer et al., 2022). At the eastern end of the profile, the upper mantle type beneath the COTZ is interpreted as serpentinized mantle based on the low velocities (Merino et al., 2021). Beneath the rafted continental block, beyond the zone of exhumed mantle, the upper mantle is argued by Merino

et al. (2021) to be faster and more homogeneous than at the eastern end of the profile, consistent with unaltered mantle. Oceanward, mantle velocities decrease gradually from the thicker, to thinner, to normal oceanic crust, although Merino et al. (2021) make no mention of possible serpentinization so the unaltered mantle type is extended to the western limit of the profile in this synthesis. The COTZ, as defined for this synthesis, extends for 260 km, and includes the orphaned continental block (Table 2).

## 5.5   Nova Scotia-Morocco margins

The Nova Scotia and Moroccan conjugate margin pair were the first to experience rifting during the late Triassic and subsequent breakup during the Jurassic within the study region (Schettino and Turco, 2009; Sibuet et al., 2012), with Morocco acting as an independent small plate detached from mainland Africa (Beauchamp et al., 1996; Piqué et al., 1998; Labails et al., 2010). Three RWAR surveys on the Nova Scotia margin (profiles 15, 17, and 19) are presented here, with the oceanwardmost half of

365 profile 15 covered by both a forward modelled ray traced velocity model (model 15a in Fig. 6; Lau et al. (2018)) as well as a Full-Waveform Inversion model (model 15b in Fig. 6; Jian et al. (2021)) derived using the same OBS data. On the Moroccan side, two RWAR surveys (profiles 16 and 18 in Fig. 1) are used for comparison, corresponding to the SISMAR 4 (Contrucci et al., 2004) and MIRROR 1 (Biari et al., 2015) surveys.

    The OETR-2009 RWAR survey (profiles 15a and 15b in this synthesis) followed the track of a coincident seismic reflection

profile, GXT-2000, with OBS spaced between less than 3 km (over the COTZ) and more than 10 km apart (Lau et al., 2018; Jian et al., 2021). For OBS velocity modelling along profile 15a, boundaries on the seismic reflection profile were used to define sedimentary layer geometries. Gravity modelling was performed following velocity modelling for validation (Lau et al., 2018). Meanwhile, the model for profile 15b was obtained using Full Waveform Inversion (Jian et al., 2021) on a subset of the OBS data and no other *a priori* information. Using the FWI-generated velocity model, pre-stack depth migration of the

coincident MCS data was performed and used to constrain the interpretation (Fig. 6).

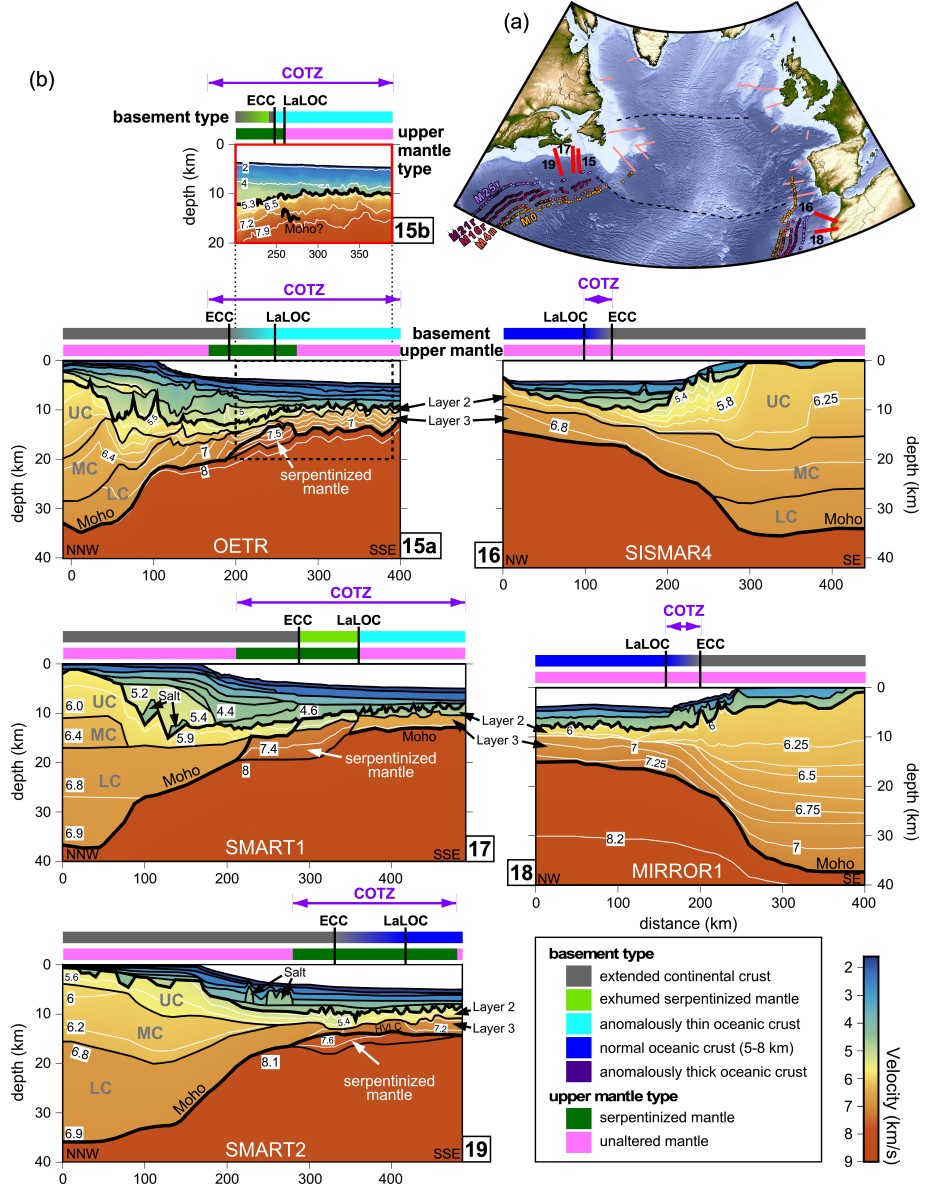

**Figure 6.** (a) Topography/bathymetry map with location of velocity models in (b) highlighted in red, and all other considered velocity model locations in this study shown in pink. (b) Velocity models reproduced from 15a: Lau et al. (2018), 15b: Jian et al. (2021), 16: Contrucci et al. (2004), 17: Funck et al. (2004), 18: Biari et al. (2015), and 19: Wu et al. (2006). In (a), the dashed black lines are select fracture zones and magnetic chron anomalies from Seton et al. (2014) are plotted as coloured circles (M25r, M21r, M16r, M4n, M0). In (b), model boundaries are shown as solid black lines and velocity contours are shown as solid white lines within the crust. Interpreted basement types and upper mantle types are plotted above each velocity model with colours explained in the legend. Abbreviations: COTZ, continent-ocean transition zone; ECC, edge of continental crust; LaLOC, landward limit of oceanic crust; LC, lower crust; MC, middle crust; UC, upper crust.

The SMART 1 (Funck et al., 2004) and SMART 2 (Wu et al., 2006) RWAR lines (profiles 17 and 19 in Fig. 1, respectively) were acquired in 2001 and coincident seismic reflection data (Keen et al., 1991; Keen and Potter, 1995) were used to define the shallow model layer geometries. Salt bodies were explicitly placed within the models based on their identification along the MCS profiles. Gravity modelling was performed for both velocity models to support the interpretations.

For all three long Nova Scotian profiles (15a, 17, and 19 in Fig. 6), gradual continental necking zones are observed, thinning from 30 km to less than 10 km over a lateral distance of approximately 200 km. The two southern profiles (SMART 1 and SMART 2; profiles 17 and 19 in Fig. 6) reveal thinned continental crust beyond the necking zone extending 60 to 80 km oceanward over serpentinized mantle. Meanwhile, the OETR profile (profile 15a in Fig. 6) has a more restricted extent of thinned continental crust beyond the necking zone, with most of it overlying unaltered mantle.

In terms of exhumation of serpentinized mantle, profiles 15b and 17 are interpreted by Jian et al. (2021) and by Funck et al. (2004) as comprising <30 and 80 km-wide zones of exhumed serpentinized mantle, respectively, while profile 15a does not show evidence of mantle exhumation at all. Instead, profile 15a shows a gradual transition from thinned continental to embryonic thin oceanic crust across the COTZ (Fig. 6). Meanwhile, the southernmost SMART 2 line (profile 19 in Fig. 6) shows thinned continental crust transitioning to normal oceanic crust.

North of profile 19 (SMART 2 line), which encompasses normal 5 km-thick oceanic crust, anomalously thin oceanic crust of less than 5 km thickness is modelled for profiles 15a (Lau et al., 2018), 15b (Jian et al., 2021), and 17 (Funck et al., 2004). These are all attributed to low melt supply during the initiation of seafloor spreading. By including anomalously thin oceanic crust in the COTZ definition herein, the resulting COTZ lateral extents are 235 km (profile 15a), 225 km (profile 15b, adopting the same landward limit as 15a), and 280 km (profile 17). The lateral extent of the COTZ for profile 19 is only slightly narrower

at 200 km but that width is based on the observed extent of unexhumed serpentinized mantle in the velocity model from Wu et al. (2006).

    Two Moroccan profiles (SISMAR 4 and MIRROR 1, profiles 16 and 18 in Fig. 6) are included in this synthesis. Both profiles involved the combined acquisition of both MCS and RWAR data, with the former being used to constrain the shallow layer geometries for the latter. The velocity model interpretations were supported by gravity modelling. The Moroccan velocity

models show similar structures, with slightly more abrupt continental necking zones compared to the conjugate Nova Scotian profiles, with crustal thinning from 30 km to less than 10 km occurring over a lateral distance of approximately 160 km. Both Moroccan models comprise very narrow 30 to 40 km-wide COTZ with extended continental crust transitioning to normal ocean crust. There is no evidence of serpentinized mantle or of exhumed mantle in the basement along the Moroccan margin based on the RWAR models generated to date (Biari et al., 2015; Klingelhoefer et al., 2016).

**6   Discussion**

The basement types and upper mantle types interpreted in the velocity models presented in Figs. 2 to 6 are summarized in the maps presented in Fig. 7 and in the diagram in Fig. 8. The map views allow for easier regional interpretations and correlations,

while the diagram allows for side by side and conjugate margin comparisons. The widths of the individual basement and upper mantle components, as well as the interpreted COTZ along each profile, are also listed in Table 2.

For the interpreted basement types (Fig. 7a, and Fig. 8), the greatest variations are observed with regards to the presence/absence, and extent if present, of exhumed serpentinized mantle, and with regards to the nature of the oceanic crust (thin versus thick versus normal). The widths of these respective zones are listed in the first two columns of Table 2.

On the North American margins (Fig. 8, left side), exhumed serpentinized mantle is modelled on all of the profiles other than profile 9 across Flemish Cap, profile 15a offshore northern Nova Scotia, and profile 19 along the southern Nova Scotian

margin. The widths of the exhumed zones vary from 20 km (profile 6) to 95 km (profile 13). On the European margins (Fig. 8, right side), south of the northern Irish margin, which lacks RWAR constraints, exhumed serpentinized mantle is ubiquitous down through Iberia, ranging in width from 10 km (profile 10) to 160 km (profile 12), with the narrowest zone outboard of the Galicia Bank continental ribbon. Offshore Morocco, no exhumed serpentinized mantle has been observed on RWAR models published to date, revealing a fundamental asymmetry with its Nova Scotian conjugate margin pair (Fig. 8).

The nature of the oceanic crust outboard of the North American margins varies between anomalously thin oceanic crust (profiles 4, 9, 15a, 15b, and 17 in Fig. 7a) and normal oceanic crust (profiles 1, 6, 13, and 19 in Fig. 7a). The densest concentration of anomalously thin oceanic crust is outboard of northern Nova Scotia but elsewhere the variability seems random, perhaps reflecting variable seafloor spreading processes acting at the local scale or simply modelling variations (e.g., OBS spacing, modelling methodology). On the European margins, only two models (profiles 10 and 14) reveal zones of anomalously thin

oceanic crust, with the latter model also exhibiting a 60 km-wide zone of anomalously thick oceanic crust immediately inboard of the thin oceanic crust. Merino et al. (2021) interpret this significant lateral variation as corresponding to the J anomaly.

For the interpreted upper mantle types (Fig. 7b and Fig. 8), sub-crustal serpentinized mantle appears ubiquitous to all but the Moroccan profiles investigated in this synthesis, with significant variations in terms of the lateral extent of serpentinized mantle within the COTZ of individual models, from 55 km (profile 10) to 230 km (profile 14). In addition, wherever failed rifts

have been interpreted in the presented profiles (Orphan Basin, Rockall Basin, Porcupine Basin, Galicia Interior Basin; profiles 3, 4, 5, and 11 in Fig. 7b; see Table 2), serpentinized mantle is always detected, varying in width from 70 km (profile 11) to 260 km (profile 3). It should be noted though that another RWAR profile across the Orphan Basin but away from the COTZ (not shown; Lau et al. (2015)), oblique to profile 4, did not show evidence for mantle serpentinization despite revealing failed rifts.

Regardless of whether mantle exhumation is achieved or not, serpentinization of mantle requires appropriate temperatures (Bonatti et al., 1984) and that any remaining crust is completely embrittled (Pérez-Gussinyé and Reston, 2001), such that throughgoing crustal faults transport fluids into the mantle. This process can involve an extended period of ultraslow rifting (<20mm/yr; Sauter et al. (2011, 2018); Grevemeyer et al. (2018)), in which sufficient cooling of the lithosphere leads to embrittlement and mantle decompression melting is suppressed. The widespread distribution of serpentinized mantle through-

out the study region (other than for the Moroccan margin) may be consistent with lengthy ultraslow rifting for the early stages of the southern North Atlantic opening.

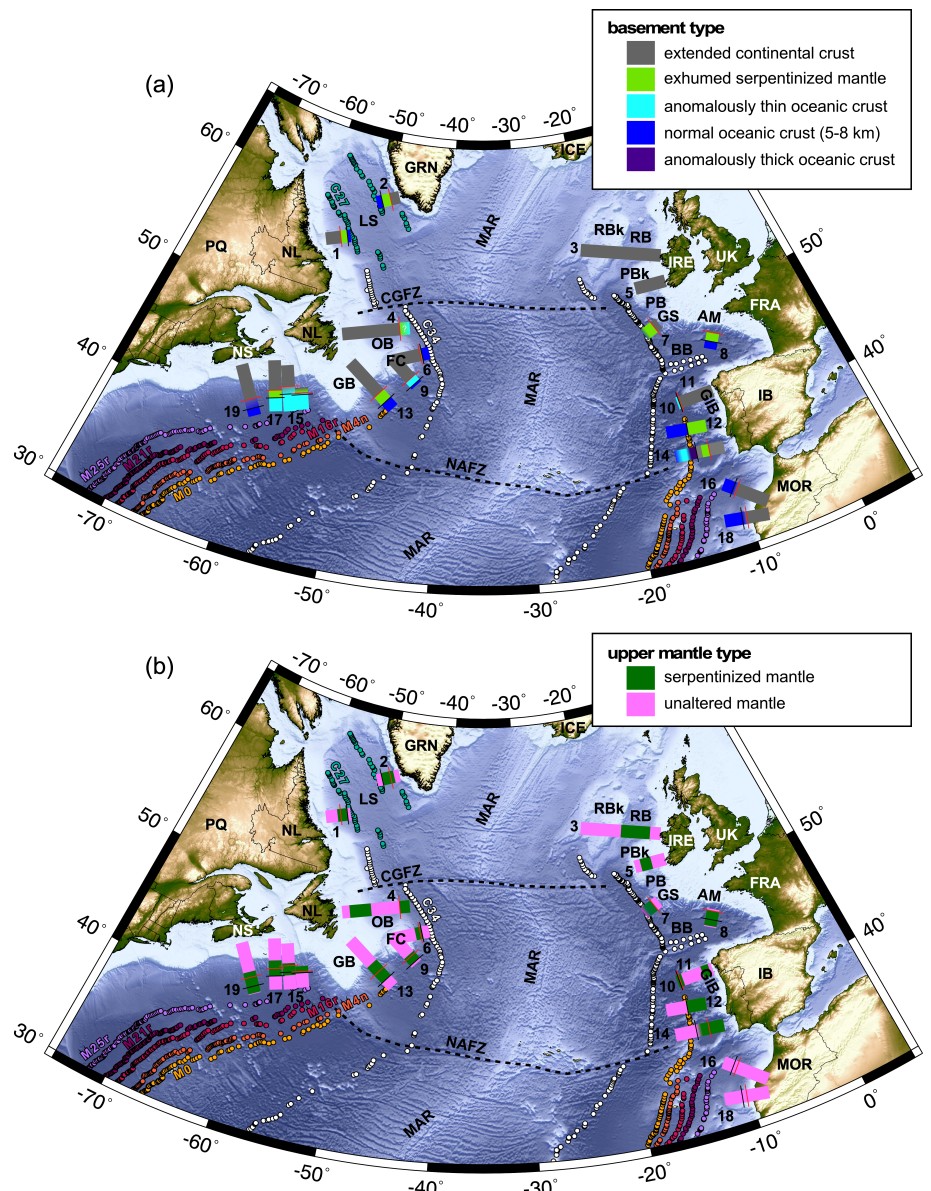

**Figure 7.** Topographic/bathymetric maps from ETOPO1 (Amante and Eakins, 2009) of the study area of the central and north Atlantic Ocean, with (a) showing the interpreted basement types summarized in this work, and (b) showing the interpreted upper mantle types summarized in this work. The numbered interpretations in both plots correspond to the seismic refraction/wide-angle reflection (RWAR) profiles described in Table 1. The solid red and black lines intersecting the interpreted polygons correspond to the ECC (edge of continental crust) and LaLOC (landward limit of oceanic crust) boundaries, respectively, highlighted in Figs. 2 to 6. Select fracture zones are delimited by dashed black lines. Select magnetic chron anomalies from Seton et al. (2014) are plotted as coloured circles (M25r, M21r, M16r, M4n, M0, C34, C27) on both plots. Abbreviations: AM, Armorican Margin; BB, Bay of Biscay; CGFZ, Charlie-Gibbs Fracture Zone; FC, Flemish Cap; GB, Grand Banks; GIB, Galicia Interior Basin; GRN, Greenland; GS, Goban Spur; IB, Iberia; IRE, Ireland; LS, Labrador Sea; MAR, Mid-Atlantic Ridge; MOR, Morocco; NAFZ, Newfoundland-Azores Fracture Zone; NL, Newfoundland & Labrador; NS, Nova Scotia; OB, Orphan Basin; PB, Porcupine Basin; PBk, Porcupine Bank; RB, Rockall Basin; RBk, Rockall Bank; UK, United Kingdom.

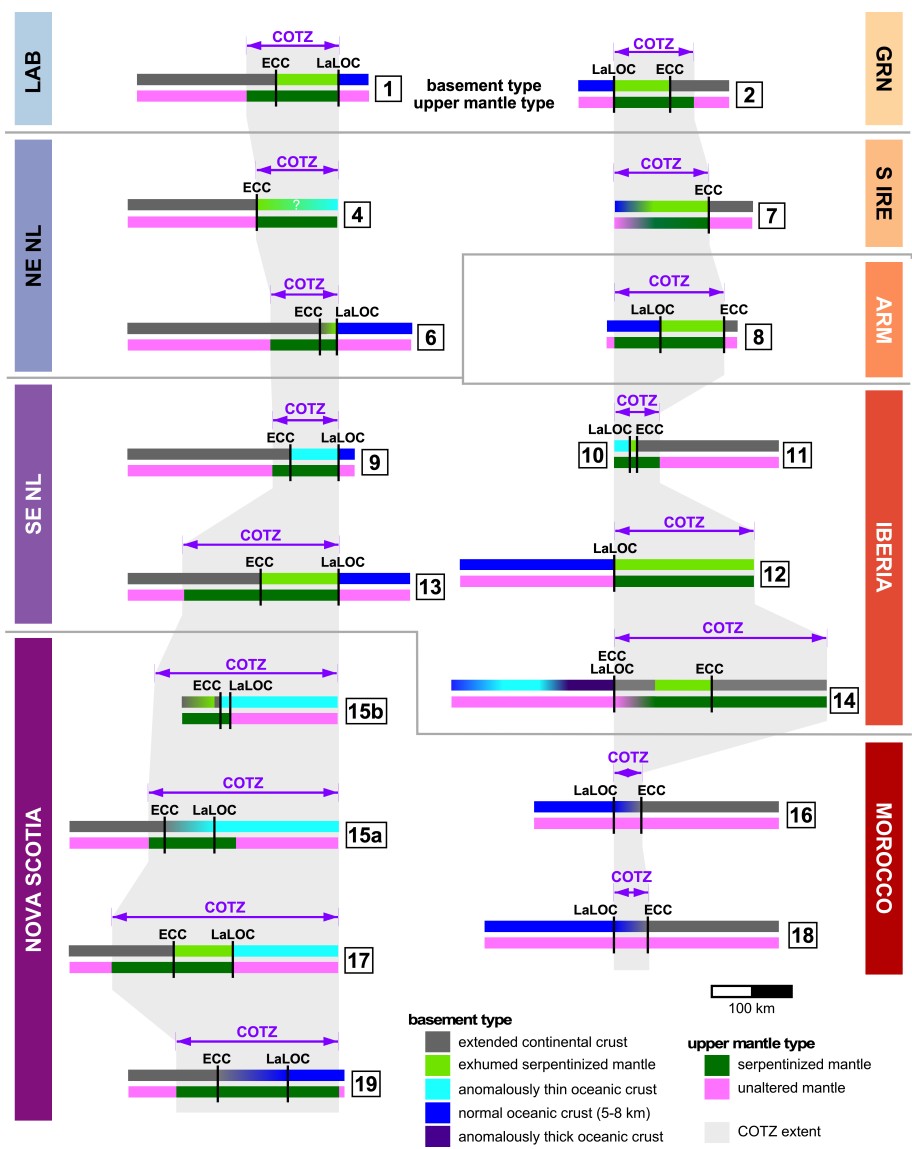

**Figure 8.** Summary diagram capturing the continent-ocean transition zone (COTZ) variations along the North American margins (left side; purple tone labels) and the European/Morrocan margins (right side; orange tone labels). For each side of the diagram, the models are aligned based on the oceanward limit of their interpreted COTZ. Landward portions of some of the sections have been truncated to conserve space. Conjugate margin pairs are roughly delimited by the gray horizontal lines. Basement types are plotted above and upper mantle types are plotted below for each profile (labelled by the numbered squares). The lateral extents of the COTZ synthesized in this contribution are plotted as grey polygons behind the simplified model interpretations. Abbreviations: ARM, Armorican Margin; ECC, edge of continental crust; GRN, Greenland; LaLOC, landward limit of oceanic crust; LAB, Labrador; NE NL, northeast Newfoundland; SE NL, southeast Newfoundland; S IRE, southern Ireland.

## 6.1 Basement, upper mantle, and COTZ regional trends: the role of inheritance

Figures 7 and 8 ultimately reveal the challenges associated with defining continent-ocean transition zones (COTZ) even within the magma-poor margin end member category. Basement types (Figs. 7a and 8) show a range of transition zone components (e.g., thinned continental crust, exhumed serpentinized mantle, anomalously thin/thick oceanic crust) with variable widths, often with conflicting interpretations in close proximity to each other. Complicating matters further is the fact that these margins often comprise extensive zones of sub-crustal and unexhumed serpentinized mantle, that extend laterally beyond the interpreted ECC and LaLOC, which are defined solely based on basement types (Fig. 7b). Furthermore, unexhumed serpentinized mantle is often found far inboard of the present day COTZ due to the presence of failed rifts (Fig. 7a).

By extrapolating the regional extents of modelled zones of exhumed serpentinized mantle, as well as the broader serpentinized mantle zones that underlie thinned continental crust and occasionally oceanic crust according to RWAR surveys, the interplay between inheritance and serpentinization can be investigated (Fig. 9), subject to the variable RWAR profile coverage. To first order, the resulting map highlights that, while inferred mantle serpentinization is pervasive on all of the interrogated margins in this synthesis (other than the Moroccan margin; Klingelhoefer et al. (2016)), exhumation of that serpentinized mantle only occurs in proximity to the locus of eventual seafloor spreading. There is also an indication, albeit biased by the available RWAR coverage, that exhumation is greatest in proximity to major fracture zones (NAFZ and CGFZ in Fig. 9), with diminished exhumation towards the Bay of Biscay extinct triple point, at least for Flemish Cap and the northwest Iberian margin.

In terms of a margin's propensity for mantle exhumation, there is no direct spatial correlation between the mapped exhumation patterns in this synthesis and particular basement terranes (Fig. 9). This signifies that while the proximal and necking rifting processes might vary according to inherited structures and crustal fabrics, as argued by Manatschal et al. (2015) and modelled by Jammes and Lavier (2019), hyperextension, mantle serpentinization and exhumation, which occur at the later stages of rifting, seem to be less influenced by the nature of their continental basement but rather by weakening through mineral hydration (Manatschal et al., 2015), by formation of detachment faults (Jammes and Lavier, 2019), due to insufficient magma supply during continued extension, and/or according to fundamental rifting parameters such as rift velocity (Huismans and Beaumont, 2007; Tetreault and Buiter, 2018). According to Brune et al. (2016), rifting velocities tend to be slow for the first two thirds of the rift evolution and then speed up for the last third, with the fast phase starting approximately 10 Myr before breakup is achieved. This piece-wise acceleration, which is thought to result from a feedback loop where strain softening processes increase non-linearly over time as deformation continues, may be sufficient to explain why unexhumed mantle serpentinization is so widespread, presumably occuring during slower rifting even within failed rifts, but that the exhumation requires the rift acceleration to breach the embrittled crust entirely, and so is only observed immediately prior to breakup (Huismans and Beaumont, 2007). For this hypothesis to extend to failed rifts like the Porcupine Basin, where mantle exhumation has been posited (Reston et al., 2001), breakup would have to have been aborted soon after exhumation, consistent with Chen et al. (2018) who interpret limited seafloor spreading in the basin. Such a scenario cannot be ruled out given the complex

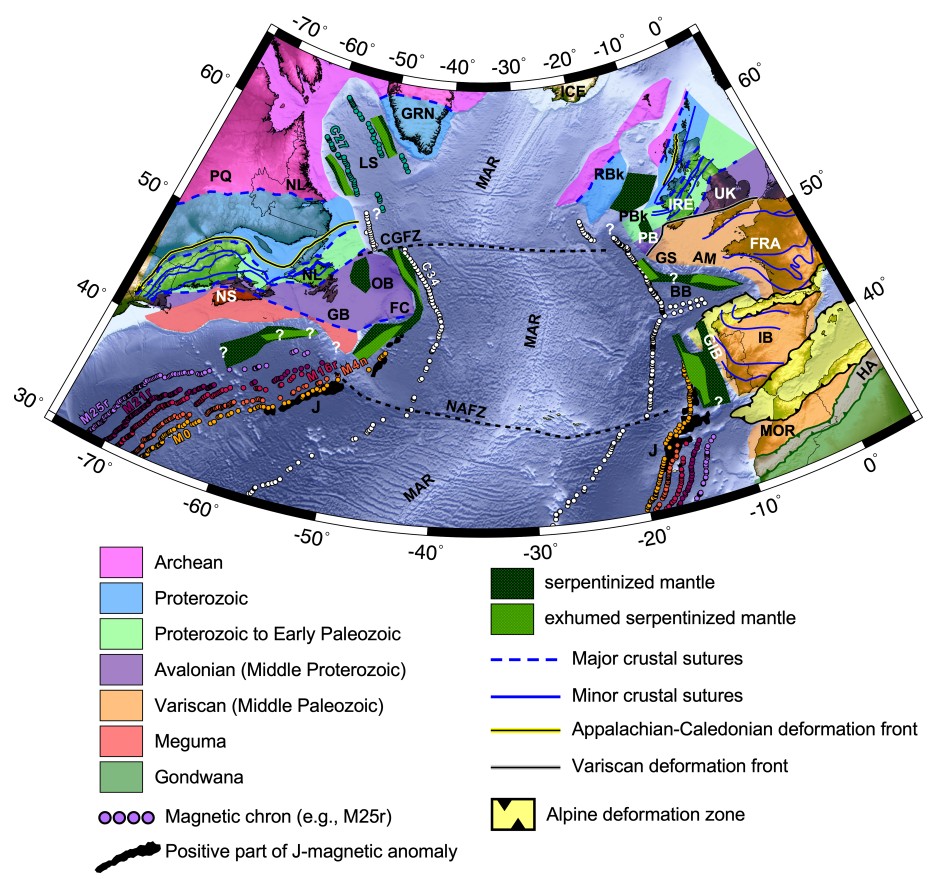

**Figure 9.** Topographic/bathymetric map from ETOPO1 (Amante and Eakins, 2009) of the study area of the southern North Atlantic Ocean subdivided by inferred basement affinity of continental crust, adapted from Jolivet et al. (2021), Tyrrell et al. (2007), Waldron et al. (2022), and Ziegler and Dèzes (2006). Interpreted extents of serpentinized mantle (hashed green polygons) with zones of exhumed serpentinized mantle (hashed light green polygons), extrapolated from Fig. 7, are overlain (areas marked with white question marks lack RWAR coverage). Select fracture zones are delimited by dashed black lines. Select magnetic chron anomalies from Seton et al. (2014) are plotted as coloured circles (M25r, M21r, M16r, M4n, M0, C34, C27). The positive portions of the J-magnetic anomaly from Nirrengarten et al. (2016) are plotted as black polygons. Abbreviations: AM, Armorican Margin; BB, Bay of Biscay; CGFZ, Charlie-Gibbs Fracture Zone; FC, Flemish Cap; GB, Grand Banks; GIB, Galicia Interior Basin; GRN, Greenland; GS, Goban Spur; HA, High Atlas; IB, Iberia; IRE, Ireland; LS, Labrador Sea; MAR, Mid-Atlantic Ridge; MOR, Morocco; NAFZ, Newfoundland-Azores Fracture Zone; NL, Newfoundland & Labrador; NS, Nova Scotia; OB, Orphan Basin; PB, Porcupine Basin; PBk, Porcupine Bank; RB, Rockall Basin; RBk, Rockall Bank; UK, United Kingdom.

3-D nature of the tectonic forcings involved with oblique rifting of lithosphere (Brune et al., 2018) with a recent history of orogenesis (Yang et al., 2021; Péron-Pinvidic et al., 2022).

In this synthesis, an attempt has been made to standardize the definition of the COTZ for the southern North Atlantic Ocean so that the COTZ lateral extents along each margin can be quantified and compared using the RWAR velocity models

considered. These interpreted COTZ extents are listed in Table 2 (fifth column) and are also shown graphically in Fig. 8 as light grey envelopes underlying the simplified model interpretations. That plot generally reveals symmetric COTZ widths all the way from the Labrador Sea to the southern Newfoundland and Iberian margins, with the Armorican margin (profile 8) perhaps departing from that symmetry due to anomalous mantle fertility and lithospheric strength than its conjugate and along-strike counterparts (Mouthereau et al., 2021). North of the Newfoundland-Azores Fracture Zone (NAFZ), the symmetry of COTZ extents occurs regardless of the basement types comprising the COTZ. However, significant asymmetry in the COTZ arises abruptly south of the NAFZ where the Nova Scotian margin COTZ is up to 9 times wider than along the Moroccan margin (consider profile 17 versus profile 16). This implies a fundamental contrast in possibly rifting mechanisms, rift velocities, thermal conditions, and/or inherited mantle rheology and fertility (Chenin et al., 2015, 2017, 2019; Mouthereau et al., 2021) across the NAFZ.

The NAFZ is a fascinating fracture zone as it aligns with the southern boundary of the present-day Grand Banks of the Newfoundland continental shelf, as well as the southern limit of the earlier St. Lawrence Promontory, which formed during the breakup of Rodinia and the creation of the Iapetus Ocean (Williams, 1984; Thomas, 2019). The spatial correlation between the abrupt change in COTZ symmetry at the NAFZ and the NAFZ possibly representing a long-lived fundamental lithospheric boundary that endured through two consecutive Wilson cycles (Thomas, 2019) underlines the point that lithospheric mantle inheritance can endure for a billion years and can supercede shallower crustal influences during subsequent tectonism (Stockmal et al., 1987; Hibbard and Waldron, 2009; Waldron et al., 2015; White and Waldron, 2022). Addressing this persistent lithospheric memory requires greater consideration of deeper processes during rifting and a recognition of the limitations associated with the simple Wilson Cycle framework.

## 6.2 Avenues for future research and potential rewards

Based on the synthesis of available RWAR velocity models of the southern North Atlantic Ocean, mapped continent-ocean transition zones (COTZ) show symmetric lateral extents north of the NAFZ and asymmetric extents further south, while comprising a range of different basement types (e.g., thinned continental crust, exhumed mantle, anomalously thin oceanic crust), ubiquitously underlain by serpentinized mantle (except for offshore Morocco). While patterns are beginning to emerge, it is abundantly clear that significant research is still needed in order to properly understand the rift-to-drift evolution along magma-poor margins. Several key avenues for future research are proposed.

Given the unmatched and crucial value of RWAR profiling for determining crustal and mantle velocities, as well as delimiting key horizontal boundaries like the Moho and lateral variations attributable to continent-ocean transitions, clearly more geophysical surveys are needed to fill coverage gaps (particularly across continent-ocean transition zones) and to ensure that margins are being compared using similar data acquisition parameters and modelling approaches. As clearly demonstrated by Grevemeyer et al. (2022), a dense spacing of OBS (10-12 km) is pivotal for high resolution characterization of continent-ocean transition zones. The resulting models allow for improved correlations with independent and co-located geophysical constraints such as shipboard magnetic data, and increase confidence in posited interpretations. Furthermore, as demonstrated by the direct comparison of models generated by conventional FMRT modelling (Lau et al., 2018) and FWI (Jian et al., 2021)

using the exact same input data but yielding different models and interpretations (models 15a and 15b in Fig. 6), comparable modelling approaches need to be used in order to reliably identify velocity model features that can be attributed to real geological phenomena. To this end, synthetic modelling of RWAR acquisition parameters to determine optimal OBS spacings and their influence on modelling outputs are needed.

Where high resolution velocity structural models do now exist, such as for southwest Iberia (Merino et al., 2021; Grevemeyer et al., 2022), significant variations in mantle exhumation and the onset of oceanic crustal accretion are revealed over relatively short spatial scales. To explain these variations, an enhanced need for numerical modelling efforts now exists, to complement existing work (Bowling and Harry, 2001; Pérez-Gussinyé et al., 2001; Pérez-Gussinyé and Reston, 2001; Pérez-Gussinyé et al., 2006; Huismans and Beaumont, 2007; Theunissen and Huismans, 2022; Liu et al., 2023), with particular emphasis placed on which key parameters (e.g., rift velocity, thermal structure, magmatic contributions, hydrothermal circulation, mantle rheology and fertility (Chenin et al., 2015, 2017, 2019; Mouthereau et al., 2021)) control and/or influence localized rifting processes and seafloor spreading mechanisms. While these parameters can be more effectively isolated and their impacts quantified using 2-D numerical models, their incorporation into 3-D models is the ultimate goal if the complexities of the southern North Atlantic are to be properly explained.

Continent-ocean transition zones hold great potential for advancing the energy transition, with serpentinized mantle representing an important global source of geologic hydrogen (Albers et al., 2021; Milkov, 2022; Liu et al., 2023), and ultramafic rocks serving the role of both effective carbon sequestration reservoirs (Goldberg et al., 2010; Schwarzenbach et al., 2013; Picazo et al., 2020; Coltat et al., 2021) as well as sources of critical minerals (Hannington et al., 2017; Patten et al., 2022). An improved understanding of the processes involved in their formation and the key controlling parameters is essential for characterizing these zones offshore and explaining their occurences in subsequent orogenesis onshore.

## 7 Conclusions

In this study, the continent-ocean transition zones (COTZ) of the southern North Atlantic Ocean have been synthesized using seismic refraction/wide-angle reflection (RWAR) profiling in order to elucidate variations in both interpreted basement types and upper mantle types, as well as to extract regional trends in COTZ components and distributions. The main findings are:

- COTZ in the southern North Atlantic Ocean are generally associated with zones of exhumed serpentinized mantle, of variable width, with most margins (other than Morocco) comprising extensive zones of unexhumed sub-crustal serpentinized mantle.

- Variations in oceanic crustal thicknesses within COTZ may reflect localized seafloor spreading processes or may simply show that different OBS spacings and methodological approaches can generate conflicting results at the oceanward limits of the RWAR profiles.

- Regardless of their basement components (e.g., thin continental, exhumed mantle, anomalously thin oceanic), COTZ of the southern North Atlantic show symmetric lateral extents north of the Newfoundland-Azores Fracture Zone all the

way to the Labrador Sea but a stark asymmetry exists further south, possibly due to fundamental differences in rifting mechanisms, rifting velocities, the thermal regime, and/or rheological and petrological variations of the continental lithosphere.

- Significant RWAR data gaps exist, particularly for the northern Irish Atlantic margins, and a clearer understanding of optimal data acquisition and modelling approaches is needed to properly characterize continent-ocean transition zones using these methods.

- A more profound understanding of continent-ocean transition zones offshore is needed to identify the key parameters that drive their evolution. This groundwork is essential for evaluating the incorporation of these zones into later orogenesis revealed onshore.

*Author contributions.*  JKW developed the concept for the contribution. JKW prepared the figures and the manuscript.

*Competing interests.*  The author declares that they have no conflict of interest.

*Acknowledgements.*  This work is supported by a Natural Sciences and Engineering Research Council of Canada Discovery Grant to JKW. Marta Pérez-Gussinyé is thanked for many constructive discussions that ultimately inspired this contribution. Thomas Funck, Ingo Grevemeyer, Helen Lau, Manel Prada, and Cesar Ranero are thanked for providing their original velocity model files (see Table 1). Thomas Funck is further thanked for providing the velocity model colour palette, originally developed by Frauke Klingelhoefer, for consistency with other nearby compilations (Funck et al., 2017; Keen et al., 2022). Frédéric Mouthereau, Georgios-Pavlos Farangitakis, and two anonymous reviewers are acknowledged for their insightful and constructive comments that helped improve the manuscript.

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

**Table 1.** Labelled refraction lines in Fig. 1, the source of the velocity models presented herein, and the type of modelling used to create the model (FMRT, forward model ray tracing; TTI, tomographic traveltime inversion; FWI, full waveform inversion). The final two columns capture additional data types used to help constrain the OBS data modelling, and the independent datasets used to confirm/validate the interpreted velocity model, respectively. Abbreviations in these two last columns: DSD, deep sea drilling; G, gravity data; M, magnetic data; MCS, multi-channel seismic reflection data; SCS, single-channel seismic reflection data; SMI, seismic mirror imaging (Grion et al., 2007).

| No. | Survey line name | Citation | Source | Mod. Type | Orig. Mod. Const. | Interp. support |
|---|---|---|---|---|---|---|
| 1 | 90R1 | Chian et al. (1995) | Digitized | FMRT | MCS | G, M |
| 2 | 88R2 | Chian and Louden (1994) | Digitized | FMRT | SCS | G, M |
| 3 | RAPIDS 1 | Funck et al. (2017) | Digitized | FMRT | - | G |
| 4 | SIGNAL 1 | Welford et al. (2020) | Original file | FMRT | MCS | G |
| 5 | P03 | Prada et al. (2017) | Original file | TTI | - | G, MCS |
| 6 | FLAME 1 | Gerlings et al. (2011) | Digitized | FMRT | MCS | G |
| 7 | - | Bullock and Minshull (2005) | Digitized | FMRT | DSD, MCS | G, M |
| 8 | Norgasis 14 | Thinon et al. (2003) | Digitized | FMRT | MCS | G |
| 9 | SCREECH 1 | Funck et al. (2003) | Original file | FMRT | MCS | G |
| 10 | Réframarge Line 6 | Druet et al. (2017) | Digitized | FMRT | MCS | G, M, MCS |
| 11 | Line 17 | Pérez-Gussinyé et al. (2003) | Digitized | FMRT | MCS | MCS |
| 12 | FRAME-p3 | Grevemeyer et al. (2022) | Original file | TTI | SMI | G |
| 13 | SCREECH 3 | Lau et al. (2006a) | Digitized | FMRT | MCS | G, MCS |
| 14 | FRAME-2 | Merino et al. (2021) | Original file | TTI | MCS | G, MCS |
| 15a | OETR-2009 | Lau et al. (2018) | Original file | FMRT | MCS | G, MCS |
| 15b | OETR-2009 | Jian et al. (2021) | Digitized | FWI | - | MCS |
| 16 | SISMAR 4 | Contrucci et al. (2004) | Digitized | FMRT | MCS | G, MCS |
| 17 | SMART 1 | Funck et al. (2004) | Digitized | FMRT | MCS | G, MCS |
| 18 | MIRROR 1 | Biari et al. (2015) | Digitized | FMRT | MCS | G, MCS |
| 19 | SMART 2 | Wu et al. (2006) | Digitized | FMRT | MCS | G, MCS |

**Table 2.** Labelled refraction lines in Fig. 1, width of exhumed serpentinized mantle zone, width of anomalous oceanic crust zone (thin or thick), width of sub-crustal serpentinized mantle zone, and width of full COTZ (as delimited in Figs. 2 to 6).

| Label no. | Exhumed mantle | Anomalous oceanic crust | Sub-crustal serpentinized mantle | Interpreted COTZ |
|---|---|---|---|---|
| 1 | 80 km | - | 110 km (within COTZ) | 115 km |
| 2 | 70 km | - | 100 km (within COTZ) | 100 km |
| 3 | - | - | 260 km (failed rift) | - |
| 4 | 50 km ?? | 50 km ?? (thin) | 210 km (failed rift) and 100 km (within COTZ) | 100 km |
| 5 | - | - | 100 km (failed rift) | - |
| 6 | 20 km | - | 80 km (within COTZ) | 80 km |
| 7 | up to 120 km | - | up to 120 km (within COTZ) | 120 km |
| 8 | 80 km | - | 135 km (within COTZ) | 135 km |
| 9 | - | 60 km (thin) | 80 km (within COTZ) | 80 km |
| 10 | 10 km | 20 km (thin) | 55 km (within COTZ) | 55 km |
| 11 | - | - | 70 km (failed rift) | - |
| 12 | 160 km | - | 160 km (within COTZ) | 160 km |
| 13 | 95 km | - | 190 km (within COTZ) | 190 km |
| 14 | 60 km | 80 km (thin) and 60 km (thick) | 230 km (within COTZ) | 260 km |
| 15a | - | 200 km (thin) | 85 km (within COTZ) | 235 km |
| 15b | <30 km ?? | 145 km (thin, within COTZ) | 60 km (within COTZ) | 225 km |
| 16 | - | - | - | 30 km |
| 17 | 70 km | 130 km (thin) | 150 km (within COTZ) | 280 km |
| 18 | - | - | - | 40 km |
| 19 | - | - | 200 km (within COTZ) | 200 km |