# Peer review of "Magma-poor continent-ocean transition zones of the southern North Atlantic: a wide-angle seismic synthesis of a new frontier"

_EGUsphere, 2023_

## Author Response (AR1)

**Responses to reviewers:**

Note that original comments are in black and responses are in blue and in italics.

**Responses to reviewer #1:**

In the contribution "Magma-poor continent-ocean transition zones of the southern North Atlantic: a wide-angle seismic synthesis of a new frontier", J. K. Welford (J.K.W.) provides a synthesis of the velocity models derived from seismic refraction and wide-angle reflection profiling along the rifted margins of the southern parts of the North Atlantic plus Labrador Sea. The author uses the interpretation of these profiles to compare the along-dip extent of the domains made of thinned continental crust, exhumed and non-exhumed serpentinized mantle, and thin, normal and thick oceanic crust among the different margins, and speculates on the potential causes of their diversity among margins.

I found the manuscript very well written and referenced. It is convenient to have all the velocity models for the southern North Atlantic margins displayed with a same color scale and a same vertical exaggeration throughout the paper, and at similar vertical and horizontal scales within each figure, however I think that this contribution could benefit from a certain number of precisions and clarifications.

*Thank you for your comments about the writing and the referencing.*

My main points of concern are the following:

•   It is not clear from the manuscript whether the interpreted extent of the different "basement type" and "mantle type" domains in the rifted margin seismic sections are from J.K.W. or from the authors of the original contribution.

*That is a very good point that I did not make clear in the submitted manuscript. In response to this comment, and the next, the revised manuscript has been significantly edited and expanded. In particular, I have added a methodology section where I clearly describe what the interpretations are based on, as well as how the model sections were generated. When introducing each model, I provide significantly more detail about how they were generated and what additional datasets were brought in to support the published interpretations.*

•   Given that the present contribution aims to provide a coherent synthesis, the best may be to use one and the same methodology to do all the interpretations. If that is what was done, then the methodology needs to be presented. For instance, if the extent of the exhumed mantle corresponds to a specific velocity interval, this interval must be stated. However, when I compare the Moho trend vs the velocity model plotted in profiles 12 and 14, with the Moho trend vs the velocity model in the other profiles, I suspect that the same rule was not applied in all profiles.

*Again, thank you for this important point. I have added a methodology section to make this all clearer. Effectively, the basement type interpretations were taken directly from the published interpretations of others. For the mantle type, those were also extracted from the published models, although the lateral extents had to be extracted from the models as these were rarely reported in the original works. As a unique contribution, I have attempted to standardize the COTZ definition for the southern North Atlantic so that the models can be better compared quantitatively. This is reflected in Figure 8 where the COTZ envelopes on both sides of the southern North Atlantic are now compared graphically.*

•   It is not clear how each of these "basement type" and "mantle type" domains is defined. It should be specified whether the interpreted extent of the different domains is only based on the velocity model, or whether other data like gravimetry analyses, magnetic anomaly analyses, seismic reflection images,

drilling/dredging, etc. are included. When a different approach has been used for interpreting the domains extent in the different profiles (for instance because the same data were not necessarily available everywhere), then the evidence on which the interpretation is based must be specified for each profile.

*This is an excellent point. This information has been added for each model under consideration and made far clearer in the manuscript.*

• According to Schön, 2015, Péron-Pinvidic et al., 2016; Osmundsen et al., 2016; Karner et al., 2021, amongst others, and implied by the author in l 66-67, velocity models do not bear only one solution with respect to the petrological nature of the rocks. It would be useful to further develop and discuss this aspect, especially because the sections discussed in this contribution appear to be only discussed from the seismic velocity model perspective. If that is indeed the case, then the proposed interpretations would not be very convincing, and I would recommend to at least display some sort of error bar.

*Thank you for this compelling point. This synthesis is based exclusively on velocity models from RWAR profiling, similar to the compilation from Funck et al. 2017 in the Geological Society of London, Special Publications (doi:10.1144/SP447.9). While I agree with what is proposed in this comment, I do not have access to the original data used to build these published models and so do not have any means of computing error bars. My only option is to direct readers to the original published works from which the models were extracted, as is now done in the revised manuscript.*

• The comparison and analysis of the extent of serpentinized, exhumed or non-exhumed mantle along the southern North Atlantic undertaken in the discussion section looks weak to me because of the lack of convincing arguments to accurately identify exhumed mantle domains (distinguish them from magmatic additions, for example).

*I agree that there are large uncertainties when it comes to comparing mantle domains along rifted margins. These uncertainties are due to a) the non-uniqueness of velocity-lithology relationships and b) the paucity of exhumed mantle sites confirmed through drilling, which are effectively limited to just the Iberian margin. The velocity distributions of these anomalous zones, with depth, do tend to show trends that can be used for differentiating mantle types versus magmatic addition. For one, the serpentinized mantle velocities in these zones tend to show a gradual increase with depth, getting closer to the velocities of unaltered mantle (8 km/s), as a function of the degree of serpentinization. Also, these zones do not show a distinct Moho. In contrast, zones with magmatic addition, particularly in the lower crust, will still show a Moho, revealed by PmP reflections. I have added these explanations to the text to hopefully make readers more confident in the discussion.*

Additional minor points are listed below:

L 20: Why mentioning the Wilson Cycle here? it is never discussed in the manuscript.

*Excellent point. I have now brought the Wilson Cycle concept back into the Discussion.*

L 53-54: I'd specify that wide tracts of exhumed serpentinized mantle can display variable amounts of magmatic products (e.g.: Bronner et al., 2011; Epin et al., 2019).

*Agree. Text added.*

L 59: Reston (2009) states "Magma-poor, magma-dominated and transform are all approximately equally numerous" (see caption of his fig. 1), but according to the more recent map by Haupert et al. (2016), magma-poor margins may be more common than magma-rich ones.

*Thank you for this comment. I have edited the text accordingly to just focus on the margin types of the southern North Atlantic and avoid global statements.*

L 66-67: As said in the point 4) above, I think this would merit some further development and referencing, since the present paper is about one of these geophysical methods that provide non-unique solutions. The occurrence of a "S-reflector" or the lack of Moho reflectivity on seismic reflection sections can for instance be used to support interpretations of exhumed mantle.

*I wholeheartedly agree with this comment. That being said, the distribution of published seismic reflection lines coinciding with the presented velocity models is not consistent. I have added mention of the lack of Moho when identifying exhumed mantle but have shied away from discussion of the S-reflector as it is only really relevant for the Iberian margin. The serpentinized mantle interpretations included in this synthesis have been lifted from published works, where those authors made the case to support their interpretations. For this synthesis, I have adopted their interpretations while providing additional details about the datasets that they ultimately used.*

L 113: Please explicit how the continent-ocean transition zone (COTZ) is defined: from the figures, it appears to correspond to the zone where serpentinized mantle occurs, but this is never stated in the text.

*Thank you for this comment. I have standardized the COTZ definition for this work and ensured that all figures and descriptions adhere to this definition.*

L 155: How is "transitional crust" defined in the present contribution?

*I have explicitly defined what is meant by transitional crust in the revised text.*

L 160-161: Thinon et al. (2003) rely on a gravity analysis and reflection seismic data in addition to velocity models to support their interpretation. I think this should be mentioned.

*Agree. These details have been added to the revised text.*

L 178: Please specify how "oceanic crust" is defined in this study. Is it based on the only seismic velocity profile? For instance, what is the difference between exhumed mantle with scarce magmatic additions and thin oceanic crust?

*The interpretations of oceanic crust are lifted directly from the published papers from which the velocity models were extracted, as are their thicknesses. This has been made clearer in the revised text.*

L 193-194: The interpretation by Merino et al. doesn't only rely on a seismic velocity model but also on a gravity analysis. I think this should be mentioned.

*Agree. These details have been added to the revised text.*

L 260-261: I don't think this is correct, this process requires only the remaining crust to be fully brittle, and yet the upper part of the crust is always brittle, as shown by the systematic presence of faults on seismic reflection profiles. This may well be because an "infinite" amount of less than 10 °C oceanic water constantly cools the upper part of the crust.

*Agree. I have made this clearer in the revised text.*

L 270: Another (easy to observe) definition of LaLOC was proposed by Sauter et al. (2023), who describe it as the inner edge of tabular magmatic crust (top basement and Moho are flat and parallel). How does this definition compare to the (velocity-based?) definition in the present manuscript?

*This is an excellent point. I have formally defined the LaLOC and referred to Sauter et al. (2023) in the revised manuscript. As mentioned, the interpretation of oceanic crust is lifted directly from the published papers from which the velocity models were extracted. Generally, those published interpretations do agree well with the Sauter et al. (2023) definition although there are challenges at the ends of the RWAR lines where ray coverage can diminish significantly.*

L 282-283: This is inaccurate: Manatschal et al. (2015) state that deformation in the proximal and necking domains may be controlled by their inherited, and thus highly variable among margins, rheology, while deformation in the hyperextended domain may be rather controlled by a much more uniform rheology among margins, acquired through pervasive fluid alteration.

*Agree. The revised text has been altered to ensure agreement with what is stated in Manatschal et al. (2015).*

L 285-286: An alternative explanation could simply be that mantle exhumation continues as long as extension continues and not enough magma is formed to accommodate extension.

*Agree. This alternate explanation has been added to the revised manuscript.*

L 289-291: This sounds contradictory with the statement on L 260-261, where it is stated that mantle serpentinization requires ultra-slow extension. Besides, it is not clear to me what is the logical link between the two parts of this sentence: "this piece-wise acceleration [...] may be sufficient to explain why unexhumed mantle serpentinization is so widespread". This should be clarified.

*I don't necessarily agree that these statements are contradictory. Mantle serpentinization, without exhumation, simply requires very slow extension of embrittled crust. That can happen for two thirds of the rift evolution, according to Brune et al. (2016), without any issues. The piece-wise acceleration would only contribute to rupturing the overlying embrittled crust. The wide zone of exhumed serpentinized mantle would be much longer lived, having formed while the embrittled crust is still in place. I have revised the text to make this point clearer.*

L291-292: This would imply that no "failed rift" basin would display exhumed mantle. I find this statement a bit bold because the occurrence of exhumed mantle in the southern part of the Porcupine Basin cannot be ruled out.

*Fair point. That being said, it is also possible that the Porcupine Basin, if it does indeed contain exhumed mantle, was on the brink of breakup before it stalled and breakup relocated west of the Porcupine Bank due to complex 3-D regional tectonic forcings. Chen et al. (2018) do interpret oceanic crust in the southern Porcupine Basin, consistent with the notion that breakup followed closely after potential exhumation.*

**Responses to reviewer #2:**

This paper presents a useful short review of the rifting processes involved at magma-poor continent-ocean transition zones based on geophysical data (especially seismic refraction/wide-angle reflection) across the margins of south North Atlantic (between 30°N and 65°N). The author conclude in proposing some future

research directions. It is well written and geophysical constraints presented are useful. The study is somewhat qualitative. Considering the large amount of existing numerical models on the subject, perhaps some more informations e.g. on the extension velocities can be provided. I also suggest to consider the role of inheritance in its broader sense, that is considering the lithosphere mantle evolution. This would permit the full exploitation of maps presented in Figure 1 and Figure 9. My detailed comments are below.

*Thank you for your constructive comments on the manuscript.*

*It is true that the comparisons are qualitative in nature. That being said, for this revision, significant effort has been put into expanding the details about each model and standardizing the COTZ interpretations, in response to comments from all three reviewers.*

*To your compelling point about quantifying extension velocities, this would require matching the features and geometries of specific RWAR profiles with published numerical models, or else extracting such velocities from deformable plate reconstructions. Given the variability in the showcased velocity models and the uncertainties inherent in the alternative modelling methods mentioned, I would not be comfortable attempting such a quantitative analysis at the regional scale given available constraints. This certainly represents a compelling avenue for future research however.*

*To your latter point, I agree that inheritance, particularly lithospheric mantle inheritance, likely plays an important role in rifting and how deeper processes are manifest in the shallower crust. The challenge here though is that, using wide-angle seismic methods, only the very most top of the mantle velocity structure is sampled (if at all), such that there is no information with which to interrogate lithospheric mantle structure on a sufficiently deep scale to enable discussion of how these features impact rifting, as captured in the velocity models. Again, for future research, it would be informative to use some of the RWAR models as constraints for numerical models, although this is beyond the scope of this synthesis.*

Figure 1 and Figure 9: I note that there are several inconsistencies and relationships regarding the boundary between terranes/cratons (Avalonia, Gondwana) and the deformed variscan domain in western Europe and northern Africa (France, Morocco). For instance, northern morocco (MOR) is described as Gondwana but it should be included in the Variscan domain (reworked Gondwana). Actually the continental crust south of Avalonia is peri-gondwanan. this has a strong impact on architecture and rheology of MOR prior to rifting. Actually this means that the mantle (not only the crust) has been modified and this probably the main aspect of inheritance as the mantle lithosphere thickness (the most resistant part of the lithosphere) can vary a range much larger than the crust (crust is 40 km thick on average on Earth).

*This is a very important point. Both figures 1 and 9 have been edited to show Morocco, north of the High Atlas mountains, as being Variscan on the maps. Thank you for this comment.*

In addition, the drawing of the Variscan front to the North looks fine but the Rheic suture is to the south so Avalonia should also extend to the south of the variscan front. this problem is inherent to the presentation of both deformed orogenic domains and the crustal affinity domain. You can refer to Mouthereau et al. (2021) for the mapping in this area and perhaps find useful the description of the mantle evolution in that domain.

*Yes. I agree that Avalonia lies south of the Variscan front and as you mentioned, it is challenging to show both the orogenic domains and the crustal affinities within one map without making it significantly more complicated. Ultimately, having played with a few variants, I have decided to not show the southern boundary of Avalonia on the European margin because it makes the figure too complicated and the offshore extrapolations of the boundaries are simply unknown.*

I also note that the Labrador Sea is presented to occur between Archean and Proterozoic crusts. But the lithosphere below Labrador has lost its archean root as it has been modified during Paleoproterozoic and the mantle is much younger thinned, and refertilized during the Neo-Proterozoic (Connelly et al., 2000; Tappe et al., 2006). This might be keys to explain why the rifting took this direction.

*This is a compelling point but it is not clear to me how this information can be extrapolated oceanward into this synthesis and used to explain the COTZ characteristics.*

Regarding Figure 8 which synthesises geometries I am wondering if some systematics on lengths of OCTZ can be found associated with northward propagation of rifting. Lengths are possibly decreasing in the direction of transform zones. This is a feature that is predicted by 3D modelling of rifting (Jourdon et al., 2020). In the same line of thoughts, more numbers may extracted from these geophysical models such as the velocities integrated over the lifetime of OCTZ (from initiation of rifting to the onset of spreading).

*This is a very interesting point. In the revised manuscript, the COTZ definition has been standardized for the study area and a quantitative appraisal of their geometries is undertaken, with the broad patterns now included in the revised Figure 8 as well.*

*While the COTZ patterns appear symmetric north of the COTZ, they do not show a systematic decrease in length towards fracture zones, at least certainly not for the Charlie-Gibbs Fracture Zone. For the NAFZ, there is a flip from asymmetric COTZ south of the transform and symmetric ones to the north. This has now been described in the revised discussion.*

*Extracting rift velocities from static velocity models is somewhat beyond the scope of this synthesis as it would require incorporation of numerical modelling and/or deformable plate reconstructions (and their inherent constraints and uncertainties).*

I have other comments on the way RWAR results are presented. In case of serpentinised mantle I would expect the Moho reflections to be lacking. This is not shown for lines 1 and 2 of Fig. 2, line 3 and unclear in line 4 of Fig. 3, unclear for line 8 of fig. 4, and not shown for lines 12, 14 but it does for line 9 in Fig. 5. To me the same problem holds for next Fig. 6 as I see bodies of serpentinised mantle but it is unclear if its upper or lower limit actually refers to a Moho. As this is one of point the author raised in discussion 6.2 I think this should be made clear. Note I am not a geophysicist but I think these comments might help the general reader.

*This is also a very good point. Where mantle is interpreted as exhumed, there should not be a Moho (although one could technically get a wide-angle reflection off of a serpentinization front if it is abrupt enough). The models presented in this synthesis show the model boundaries as presented in the original published works, where authors did not necessarily make this clear. This is definitely the case for the velocity models in Figure 2 from the Labrador Sea. For Figure 3, there is no issue because none of the serpentinized mantle is interpreted as exhumed. Line 8 in Figure 4 is particularly unusual because they interpret a high velocity lower crust below the Moho, when this should in fact be referred to as low velocity upper mantle. The variable character for the profiles in Figure 5 and Figure 6 are completely a function of how the models were obtained (forward modelling versus inversion). To address your point, this has been clarified in the revised manuscript both in terms of a new methodology section, with specific mention of the Moho representations, and expanded descriptions of all of the velocity models.*

Line #38-40: Labails et al. (2010) indicate 190 Ma for the age of breakup and onset of oceanic spreading. How do you integrate this study in your geological background ?

*Thank you for noting this point. I have incorporated this older breakup age into the revised geological background.*

line 715: Correct "detrital"

*Good catch. Edited.*

**Responses to reviewer #3:**

General comments: This paper presents a synthesis of the evolutional models of magma-poor continent-ocean transition zones and their corresponding rifted margins using refraction and wide-angle reflection seismic focusing on the margins of the (south) North Atlantic. The study presents a good documentation, and -most importantly- standardization of the continent to ocean transition and its relative mantle and basement domains. When methodologies or studies have displayed bias or are generally sparsely sampled, this is clearly indicated and that adds to the integrity of this review. A discussion follows in the end with regards to the origin of this observed margin differentiation, supported by 3 very descriptive summarizing figures that help drive the point home.

*Thank you for your review of the manuscript and your constructive comments.*

The manuscript is overall very well written and reads really well. I believe there are two formative changes that could benefit this work as listed below and one change with regards to the content, (followed by some minor comments split per segment of the manuscript):

• 	It is unclear to me if this is the final format of the paper, however it would benefit significantly if the figures are directly below the part of the text discussing them. I.e. when I am reading section 5.2 I need to scroll up and down 3 pages. If this is just a final proofing issue please ignore this comment.

*I have attempted to fix this in the revised manuscript for easier reading. The manuscript is formatted using LaTeX so the typesetting often must be forced to comply.*

• 	On figures 2-6 it is unclear to me whether the interpretation of the basement and upper mantle along with the profile interpretation on the velocity cross-section is done from the author or the cited authors (or if one is an interpretation and the other is a direct translation).

*This is an excellent point, also raised by another reviewer. I have added a methodology section to the revised manuscript to make this point very clear. I have also added more details about each individual model and how it was generated and interpreted. Briefly, the interpretations were inherited from the published works, with some adjustments to standardize the definition of the COTZ.*

• 	On the discussion about inheritance, the role of lower crustal to mantle inheritance (or rheological diversity) seems to not be considered enough. There are quite a few studies (i.e. Autin et al. 2013, Manatschal 2015, Heron et al. 2019, Farangitakis 2020) that indicate that the domains in a rift would be controlled by previous crustal inheritance (or mantle-straddling "seeds" even lower. Would that not in turn control the location of serpentinization?

*This is another strong point, also raised by another reviewer. I agree absolutely that mantle inheritance may be controlling a great deal of the localization of serpentinization. The challenge in this synthesis is that the models are primarily crustal models, with occasional sampling of the topmost underlying mantle. Therefore, the velocity models just don't provide enough constraints for an in-depth discussion of mantle controls. That being said, in the revised manuscript, I have expanded the discussion on the serpentinization trends, hopefully adding more insights.*

Find below a list of minor specific comments for the manuscript content:

1 Introduction:

L20-24: This paragraph feels slightly out of order (Wilson cycle mentioned after magma content in rifts and not re-visited again in the manuscript). It could be omitted or reworked to start with the rift seeds in the deformation zones and then revert to how this relates to the Wilson cycle.

*Agree. The introduction has been heavily revised and the Wilson cycle discussion brought back later in the manuscript.*

Figure 1: a) It might help to add the chron symbol & (corresponding) number in the legend (not all chrons, but an example) as it would help the reader know where to focus and what to look for.

*Agree. Very good point. This has been done for the revised manuscript.*

b) Consider adding the word "select" behind the fracture zones as well, since not all fracture zones within the map are delineated/named.

*Agree. Very good point. This has been done for all relevant figure captions in the revised manuscript.*

L25-L28: This paragraph would benefit from a slight expansion on how the review is done (maybe expand the second sentence further). Also, the first sentence is written in active voice while the second is written in passive voice

*Agree. A methodological section has been added to make the synthesis process a great deal clearer. I have also checked the text for consistent use of active voice.*

2 Geological background:

L38: Would be useful to the reader to define where the "northernmost" point sits in the map (i.e. above the NAFZ, below the NAFZ?, or between 30 and 40 degrees etc)

*Agree. Good point. I mean south of the NAFZ. I have made this clear in the revised text.*

3 Characterization of rifted margins:

L53: "manifested" instead of "manifest"

*Agree. Done.*

L58: A citation for the COTs would add consistency as done with each of the previous settings in the paragraph.

*Agree. Citations have been added.*

4 Geophysical Methods:

L85: Consider a sentence that explains the reduction bias mentioned between using FMRT vs TI methods.

*The sentence has been expanded to explain that the bias is reduced due to the use of model norms rather than subjective visual inspection.*

5 Geophysical Constraints:

L174: "both" needs to be removed.

*Agree. Done.*

L185-191 & Figure 5: The J anomaly is quite extensively mentioned in this text segment; however it is not clear where it is in any of the map or cross-sections.

*Agree. Very good point. The J-magnetic anomaly has been added to the maps (Figs. 1 and 9) and relevant cross-sections (Fig. 5).*

6 Discussion:
L261: Consider adding a representative velocity for the "ultra-slow" rifting to help clarify the relative velocity.

*Agree. An ultra-slow velocity of <20 mm/yr has been added based on the work by Sauter et al. (2011,2018) and Grevemeyer et al. (2018).*

---

## Author Response (AR2)

**Responses to reviewers:**

Note that original comments are in black and responses are in blue and in italics.

**Responses to reviewer #2: Frédéric Mouthereau**

I still have some comments since I have noticed that some of the points Kim Welford said she will address in the revised ms are not in fact.

*I would like to apologize for any confusion that arose from the responses during the last round of review as I am new to the Solid Earth procedures. I continued to work on the revision, and the corresponding rebuttal letter, after the online discussion board had been closed. In the final rebuttal letter that was submitted with the revised manuscript, I amended some of my original responses relative to the online discussion but had no means of editing the online responses once the rebuttal letter was finalized.*

Kim's answer
"1) I also note that the Labrador Sea is presented to occur between Archean and Proterozoic crusts. But the lithosphere below Labrador has lost its archean root as it has been modified during Paleoproterozoic and the mantle is much younger thinned, and refertilized during the Neo-Proterozoic (Connelly et al., 2000; Tappe et al., 2006). This might be keys to explain why the rifting took this direction.

Yes. This is a compelling point that I will incorporate into the discussion."

But I don't see this point discussed in the revised ms. Please incorporate it clearly.

*As updated in the final rebuttal letter from the last round, I commented that this is a compelling point but it was not clear to me how this information could be extrapolated oceanward into this synthesis and used to explain the COTZ characteristics. As such, I decided to leave discussion of causative mechanisms for Labrador Sea rifting out of the manuscript, particularly since incorporation of the two suggested references would necessitate an even broader discussion of onshore versus offshore controlling structures that would need to be expanded to all of the margins under consideration. Instead, I have chosen to reference the Nature Communications paper by Gouiza and Naliboff (2021) as they address the influence of inheritance in the Labrador Sea rifting segmentation offshore.*

also

Kim's answer :
"In addition, the drawing of the Variscan front to the North looks fine but the Rheic suture is to the south so Avalonia should also extend to the south of the Variscan front. this problem is inherent to the presentation of both deformed orogenic domains and the crustal affinity domain. You can refer to Mouthereau et al. (2021) for the mapping in this area and perhaps find useful the description of the mantle evolution in that domain.

Yes. I agree that Avalonia lies south of the Variscan front and as you mentioned, it is challenging to show both the orogenic domains and the crustal affinities within one map without making it significantly more complicated. I will refer to your 2021 paper for ideas on how to do this most effectively.

But I don't see reference to this paper in the revised ms.

*As stated in the final rebuttal letter from the last round, yes, I agree that Avalonia lies south of the Variscan front and as you mentioned, it is challenging to show both the orogenic domains and the crustal affinities within one map without making it significantly more complicated. Ultimately, having played with a few*

*variants, I decided to not show the southern boundary of Avalonia on the European margin because it makes the figure too complicated and the offshore extrapolations of the boundaries are simply unknown. I have however now referenced Mouthereau et al. (2021) when mentioning the importance of considering mantle rheology and fertility during rifting.*

Regarding the point line 600 in the conclusions section. I suggest to add "composition" or "petrological" and "continental" because mechanical inheritance is not restricted to rheology but to density (thermally and compositionally related) and isostatic parameters more broadly.

.... of the southern North Atlantic show symmetric lateral extents north of the Newfoundland-Azores Fracture Zone all the way to the Labrador Sea but a stark asymmetry exists further south, possibly due to fundamental differences in rifting mechanisms, rifting velocities, the thermal regime, and/or rheological and petrological variations of the continental lithosphere.

*Agree. The text has been edited as suggested.*

**Responses to reviewer #3: Georgios-Pavlos Farangitakis**

*Reviewer #3 had no further suggestions and was happy for the revised manuscript to be accepted as is. I thank the reviewer for the time that he dedicated to reviewing the revised manuscript.*

**Responses to reviewer #4: Anonymous**

General comments:
This is a very useful, well written and illustrated manuscript that merits to be published. Indeed, as pointed out by the author, such a detailed synthesis of refraction seismic lines across he COTZ was missing for the southern Northern Atlantic, despite being one of the type localities of magma-poor margins.

*I would like to thank the fourth anonymous reviewer for joining in the review process mid-stream, after one full round had already been completed.*

Overall, I do not have major comments and I think that the manuscript can be published with minor modifications. Below I have some comments that being addressed, could clarify some of the weak points of the work.

1) It would be helpful to have a paragraph focusing on the Moho in the COTZ and explain the velocity characteristics of this structure and the properties of related rocks (a figure could be very helpful)

*The previously revised manuscript does include a section of text about how the Moho was defined in the synthesis. To address this comment from reviewer #4, I have added two sentences about the velocity characteristics of the Moho in that existing section in this latest revision. Specifically:*

For the PmP reflections to be clear during RWAR profiling, the velocity contrast between the lower crust and the upper mantle must be large enough to generate a detectable reflection. Typically, the velocity contrast across the Moho corresponds to 6.9 km/s for the lower crust and to 8 km/s for unaltered mantle, with mantle velocities decreasing with increasing degrees of serpentinization.

2) The term "transitional crust" used for instance on l.139, does not really make sense, since there is no transition between continental rocks, serpentinite and basalts. While velocities and other geophysical parameters may show transitions, rocks don't, and when we talk about crust, we often refer to rocks. The

same applies to "homogeneous" oceanic crust (line 142). From a geological point of view a crust is not "homogeneous". Thus, would not use such a term

*Within the rifted margin community, the term "transitional crust" is very much a standard term used. It refers to crust that may consist of varying amounts of different crustal types (e.g., thinned continental crust interspersed with pockets of serpentinized peridotites or magmatic products). In the existing text, it is explained: "Transitional crust can refer to hyperextended continental crust, exhumed mantle, embryonic oceanic crust, and any combination thereof, making the demarcation of the limits of transition zones challenging". As such, to conform to the existing literature, I have not removed the use of this term.*

*In terms of homogeneous oceanic crust, that is fair. I have replaced "homogeneous" with "classic Penrose-type" oceanic crust and provided a reference.*

3) Definitions: Here the COTZ is defined and subdivided in three subunits (a), (b) and (c) (see lines 154 to 156). However, it is not clear how the continentward limit of the COTZ (and domain (a)) is defined. Is it the necking zone, or the coupling point? This is important, since you define a width of the COTZ; so where do you start measuring. It is also not clear how you define the limit between domains (a) and (b). Is it the ECC of Nirrengarten et al. 2018?). If so, be more explicit. And finally, the limit between domain (b) and (c) needs to be defined as well as the limit of first oceanic crust. In the present version, definitions are not clear and not applied rigorously in the Figures 3 ot 6. This is clearly a weakness of the present version of the manuscript.

*As explained in the text, I define the landward limit of the COTZ as corresponding to "hyperextended continental crust underlain by serpentinized mantle". This does not always correspond to the necking zone but it can. As you mention, the coupling point is perhaps a better definition so I have now stated that in the text explicitly. Subunit (b) only exists where mantle is interpreted to be exhumed and its landward limit would be the ECC (this has now been added to the text). Finally, the boundary between subunits (b) and (c) occurs where exhumed mantle ends and anomalously thin oceanic crust begins. While this boundary will generally correspond to the LaLOC according to the definition of Sauter et al. (2023) used herein, the oceanward limit of the COTZ may be further oceanward if the oceanic crust has not reached normal thickness, or if the unexhumed serpentinized mantle extends further oceanward than the LaLOC. I rechecked the COTZ extents shown in Figures 3 to 6 and found that all but profile 8 did conform to the standardized COTZ definition introduced here. The COTZ extent for profile 8 has now been updated in Figures 4 and 8 to conform to the new definition.*

4) It is not clear how domains of serpentinized mantle underlying thinned crust are defined. What are the criteria used?

*Serpentinized mantle underlying thinned crust is defined solely based on the presence of upper mantle velocities lower than 8 km/s. This has been made clearer in the latest revision of the manuscript.*

5) It would be useful to show the locations of drill holes penetrating basement and showing the type of rocks they recovered (this is the strongest support for your interpretation)

*For the southern North Atlantic, there are very, very few drill holes that penetrate basement in the COTZ. These are effectively limited to the Iberian margin, with only one drill hole on the Newfoundland margin. Consequently, a synthesis of COTZ across the entire southern North Atlantic can only be accomplished using geophysical datasets, such as the velocity models investigated herein. Given their limited contribution to this synthesis, and the fact that the maps are already quite busy, I have not plotted basement drill holes for this study. Where drilling information has influenced the velocity model building, it is already explained in the corresponding published works.*

6) Serpentinization is not only dependent on the embrittlement, but also depends on the temperature (should be ≤ 300 to 350°C. This may be added in line 428 and you may add a reference

*Agree. Edit and reference added as suggested (Bonatti et al., 1984).*

7) You discuss importance of rates (lines 458) during rifting (how are they determined? It's difficult for rifting, there are no time lines such as magnetic lineations!! You do not discuss importance of inherited depleted mantle. This may be useful. See work of Chenin et al.

*Agree. It is difficult to be quantitative in this respect. Rifting velocity inferences are currently unconstrained, other than from insights from numerical modelling and deformable plate reconstructions. And yes, inherited mantle fertility is important for understanding rifting processes. This has already been mentioned in the revised text but additional references have now been added (particularly to the work by Chenin and others). Unfortunately, the mantle fertility question cannot be addressed with the present synthesis because the RWAR velocity models only provide limited information about the uppermost mantle.*

8) Figures:
• Moho: would use dashed or dotted lines for places where MOHO is not defined and assumed to be a hydration front due to serpentinization. In the present version you use black lines for many other interfaces, and it is not clear how and where you put Moho and with what criteria

*This is a compelling point, although difficult to illustrate in the velocity model representations. As mentioned in the text, the Moho is defined by a thicker line where supported by wide-angle reflections in the published works. Where this is not so, the black lines are often just corresponding to model boundaries and not hydration fronts. The Moho interpretations come directly from the published works included in the synthesis and are not reassigned herein so I prefer not to alter the models as they currently appear in the literature.*

Fig. 3 is the domain of serpentinized mantle in section 3 realistic? Can serpentinization reach so deep below top basement

*The serpentinized mantle in section 3 is extracted directly from the published works and I do not have access to the supporting dataset with which to assess its plausibility. Without extra information, I cannot alter the published model.*

Fig. 4 how could you define serpentinite underneath normal oceanic crust in section 8.

*As mentioned above, unexhumed serpentinized mantle is defined solely based on the presence of upper mantle velocities lower than 8 km/s. The serpentinite interpretation for the HVLC in section 8 is extracted directly from Thinon et al., 2003.*

Fig. 5 not clear why similar velocities result in different interpretations and if Moho in section 14 is really determined

*Velocities are unfortunately not unique to specific crustal types, although velocity gradients and the regional context can help minimize uncertainties in their interpretation. As for the Moho in section 14, the PmP reflected arrivals are simultaneously inverted with the refracted arrivals by Merino et al., 2021, so yes, their Moho is geophysically determined.*

Fig. 6 the result that there is no mantle exhumation on the Moroccan margin are an interpretation, reflection

seismic data show locally evidence for mantle exhumation in the northern Moroccan margin; so would not be too dogmatic on this point.

*I agree on this point. The verbiage when discussing the Moroccan margin is specifically made non-definitive for this reason. The text in the revised manuscript stated "There is no evidence of serpentinized mantle or of exhumed mantle in the basement along the Moroccan margin based on models generated to date". I have now specified in the newly revised text that I am referring to RWAR models generated to date. As an aside, I have recently performed some gravity inversions (not yet published) which seem to suggest greater complexity than is captured in the published velocity models. Unfortunately, there are no RWAR velocity models indicating otherwise as of yet.*

Fig. 8 why not aligning along LaLOC. At the present version the COTZ goes oceanwards to the LaLOC (see lines Nova Scotia), is this not questioning the location of the COTZ or the LaLOC on these lines? Why do you don't put LaLOC in section 7 (you show a blue crust)?

*I chose to align the sections according to the oceanward limit of their COTZ extent, and not LaLOC because LaLOC is based on an interpretation of the basement as boxy oceanic crust (regardless of its thickness, as suggested by Sauter et al. (2023)). As mentioned above, the COTZ can extend further oceanward of LaLOC. For instance, for profile 19, LaLOC is landward of the extent of unexhumed serpentinized mantle. As such, the COTZ is defined according to the unexhumed serpentinized mantle rather than the LaLOC.*

*For section 7, the authors never report finding evidence of typical oceanic crust so effectively the location of LaLOC there is undefined and I show the colour as a gradient from green to blue to show that uncertainty.*